# Land use change and ecological sensitivity in the Qingdao West Coast new area: A 30-year analysis and future scenario simulation

Tong Zhou 🔴*, Jiabin Wang 🔴, Yaning Zhao, Yi Sheng

College of Civil Engineering and Architecture, Shandong University of Science and Technology, Qingdao, Shandong Province, China

* zhoutong@sdust.edu.cn

## Abstract

This study aims to reveal the long-term ecological evolution in the Qingdao West Coast New Area (QWCNA) and predict future trends to support its sustainable development. Firstly, it employed GIS-based land use dynamic indices and transfer matrix analyses to assess land use changes from 1990–2020. Secondly, this study assessed ecological sensitivity (1990–2020) using an Analytic Hierarchy Process (AHP) weighted 7-factor system covering the natural environment, land cover, and accessibility. Thirdly, the Patch-Generating Land Use Simulation (PLUS) model predicted 2030 land use under Natural Development (ND), Urban Development (UD), and Ecological Protection (EP) scenarios, which were subsequently used to evaluate future ecological sensitivity patterns. The main results indicate that a drastic land use transformation occurred between 1990 and 2020, marked by a significant expansion of construction land and forestland. This expansion primarily displaced cultivated land, grassland, water bodies, and unused land, driven by rapid urbanization. Furthermore, spatially distinct ecological sensitivity patterns evolved; lower sensitivity areas increased alongside urban expansion, while higher sensitivity zones (High and Extremely High), concentrated around the Xiaozhu, Dazhu, and Cangma–Tiejue Mts, expanded notably. The expansion of these higher sensitivity zones suggests potential environmental improvement attributed to enhanced conservation efforts. Future simulations show that the EP scenario best aligns with sustainability goals, maximizing the extent of High and Extremely High sensitivity areas by 2030 compared to the ND and UD scenarios.

## 1. Introduction

With continuous population growth, the demand for production and living space constantly increases, leading to the progressive compression of ecological space. Conflicts among urban construction, agricultural production, and ecosystem service

**Data availability statement:** Yes - all data are fully available without restriction; We have uploaded the raw data, research result data, and figures to the Figshare repository. You can access the dataset via the following link or citation: https://figshare.com/s/16f2e344582e196abbb6.

**Funding:** The author(s) received no specific funding for this work.

**Competing interests:** The authors have declared that no competing interests exist.

functions are intensifying, increasing the scope and intensity of impacts on the local natural environment and ecosystems [1–3].

Ecological sensitivity refers to the degree of an ecosystem's response to natural environmental changes and human disturbances within a region, reflecting the likelihood of ecological problems arising when the system is impacted by these factors [4–6]. Essentially, ecological sensitivity assessment is a method for identifying potential ecological problems under existing natural and anthropogenic backgrounds, providing valuable information for their prevention and management [7–9]. The evaluation of ecological sensitivity is of great significance for regional environmental protection and the rational development of natural resources [10–12]. It can guide practices such as ecological zoning, landscape planning, and urban planning, thereby laying a solid foundation for regional ecological civilization construction and the harmonious development between humanity and nature [13–16].

However, current global research on ecological sensitivity often focuses on static assessments or single-scenario predictions, lacking a comprehensive, long-term integrated framework that links historical evolution with future multi-scenario simulations. While existing research has evaluated the ecological sensitivity of the West Coast New Area [17,18], our study makes three key contributions to address these gaps. First, we conducted a comprehensive 30-year longitudinal analysis (1990–2020), capturing the region's complete transformation from administrative units to a national-level new area. Second, we utilized the advanced Patch-Generating Land Use Simulation (PLUS) model, which offers superior accuracy and deeper insights into land use change mechanisms compared to traditional models. Third, we constructed an integrated 'past-present-future' framework that directly correlates historical land use dynamics with ecological sensitivity, providing a robust scientific basis for forecasting and evaluating future scenarios. This rigorous approach offers significant value for sustainable regional development.

## 2. Study area

### 2.1. Study area description

The QWCNA, administered by Qingdao City, Shandong Province, is situated on the southwestern edge of the Shandong Peninsula, bordering Jiaozhou Bay. It faces the Yellow Sea to the south, connects with Jiaozhou City to the north, and neighbors Zhucheng City, Wulian County, and Rizhao City to the west. Its geographic extent lies between latitudes 35°35′ to 36°08′ N and longitudes 119°30′ to 120°11′ E. The area features prominent mountain systems, including Xiaozhu, Dazhu Mountain, and Cangma-Tiejue. It encompasses a land area of approximately 2088 km² and a sea area of approximately 5270 km² [19].

As the core bearing area of the Shandong Peninsula Blue Economic Zone, the QWCNA adheres to its strategic positioning and urban spatial system planning, emphasizing a balance between work and residence and adopting a spatial development model characterized by 'clusters + peripheral towns.' The core area comprises six major functional clusters: Wangtai, Huangdao, Lingshanwei (Central Business District), Jiaonan, Guzhenkou, and Dongjiakou. Six peripheral

towns—Baoshan, Liuwang, Dacun, Cangnan, Dachang, and Haiqing—surround these forming a multi-center, networked urban pattern [20] (Fig 1).

Owing to its unique geographical conditions between mountains and the sea, the West Coast New Area boasts magnificent natural scenery and harbors abundant and diverse tourism resources. The World Tourism Organization once evaluated it as one of Northern China's most promising tourism and resort destinations. Furthermore, as the ninth state-level new area approved for establishment by the State Council, the West Coast New Area has also achieved remarkable accomplishments in ecological civilization construction. Between 2004 and 2011, it successfully established beautiful national-level ecological towns. It was awarded the "National Ecological Zone" title in 2016, reflecting its efforts and effectiveness in coordinating economic development with environmental protection [21].

## 3. Research methods

### 3.1. Data sources and preprocessing

The multi-source datasets utilized in this study primarily cover natural environmental factors (e.g., DEM, climate) and socio-economic drivers (e.g., population, GDP), as detailed in Table 1. This study relies exclusively on publicly accessible open-source datasets; as no fieldwork was conducted and no proprietary data were used, specific permits were not required.

All raw data were preprocessed using ArcGIS 10.8, involving projection transformation, clipping, resampling, and reclassification. A unified coordinate system (WGS_1984_UTM_Zone_50N) was adopted, and all datasets were resampled to a spatial resolution of 30 m × 30 m. Outliers were corrected via interpolation, and driving factors were rasterized to ensure spatial consistency with land use data [22]. Regarding classification, this study adhered to the National Standard (GB/T 21010−2017) to categorize land use into six primary types [23]: cultivated land, forest land, grassland, water bodies, construction land, and unused land (Fig 2).

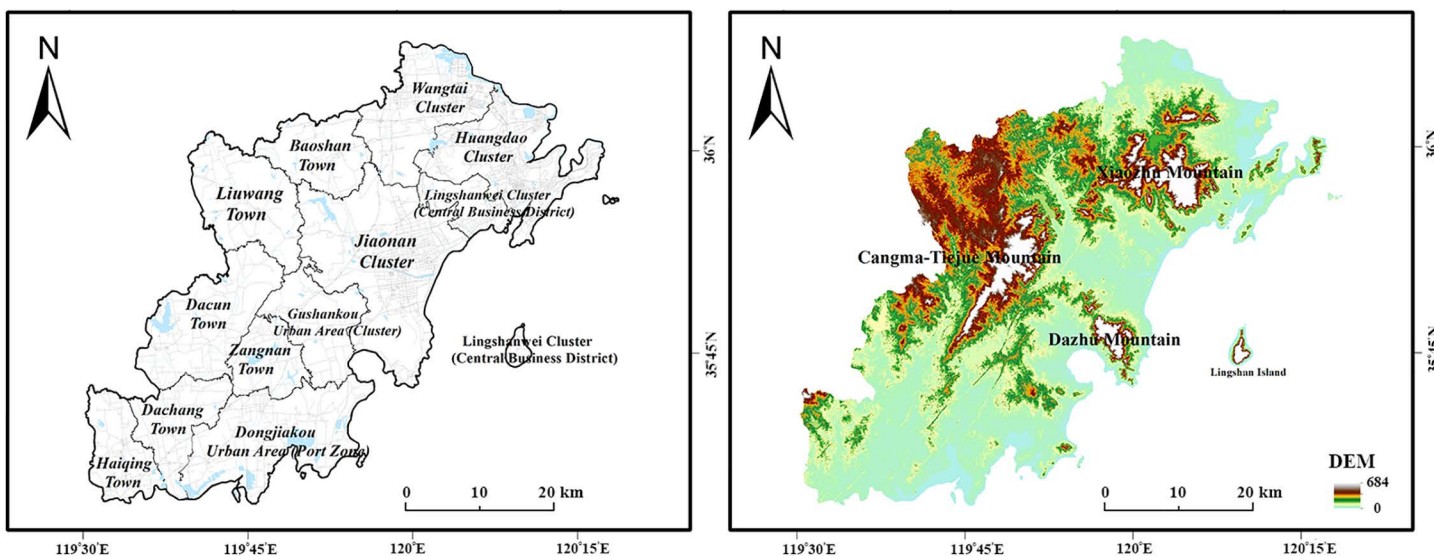

**Fig 1. Schematic diagram of the study area.**

**Table 1. The sources of data.**

| Name | Source | Website/URL |
|---|---|---|
| Base Map of the Study Area | National Platform for Common GeoSpatial Information Services | https://www.tianditu.gov.cn/ |
| Population | Geospatial Platform | http://gis5g.com/ |
| Elevation(DEM) | Geospatial Data Cloud | http://www.gscloud.cn |
| Slope | DEM | – |
| Aspect | DEM | – |
| Water Body Buffer | National Catalogue Service for Geographic Information | https://www.webmap.cn |
| Road Vector Data | Open Street Map | https://openmaptiles.org/ |
| Annual Average Precipitation | Geospatial Platform | http://gis5g.com/ |
| Land Cover | Resource and Environmental Science Data Platform | https://www.resdc.cn/ |
| NDVI | National Tibetan Plateau Data Center | https://data.tpdc.ac.cn |
| Annual Average Temperature | Geospatial Platform | http://gis5g.com/ |
| Nighttime Light Data | Resource and Environmental Science Data Platform | https://www.resdc.cn// |
| GDP | Resource and Environment Science and Data Center | https://www.resdc.cn/ |

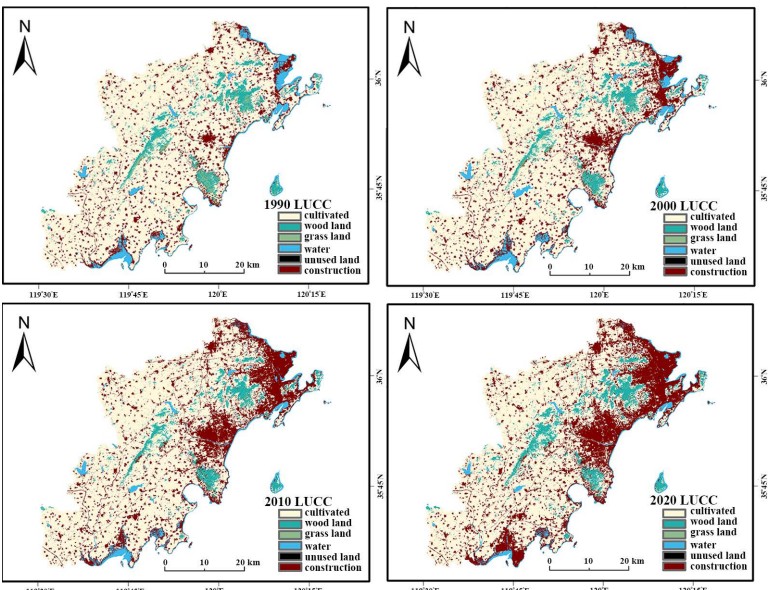

**Fig 2. Land use changes in the study area from 1990 to 2020.**

### 3.2. Research framework

To systematically reveal the mechanism of ecological evolution and predict future trends, this study established a comprehensive research framework as illustrated in Fig 3. The methodological workflow comprises three distinct stages:

1) Historical Spatio-temporal Analysis (1990–2020): Utilizing land use datasets from four periods, we analyzed historical transitions via GIS and transfer matrix methods.

2) Ecological Sensitivity Assessment: We constructed a multi-factor evaluation system weighted by AHP to assess the spatio-temporal evolution of ecological sensitivity.

**Fig 3. Study method flowchart.**

3) Future Scenario Simulation & Assessment (2030): Integrating regional planning policies, we established three development scenarios (Natural Development, Urban Development, and Ecological Protection). The PLUS model was employed to simulate land use patterns for 2030, which subsequently served as the core basis for prospectively assessing future ecological sensitivity.

### 3.3. Spatio-temporal evolution of land use

To understand the temporal changes in land use forms, composition, and area, we conducted a time-series analysis of the West Coast New Area.

**3.3.1. Single land use dynamic index.** First, the single land use dynamic index was analyzed. The Index refers to the change in area of a specific land use type within a region over a certain period. It intuitively reflects the magnitude and speed of change for various land use types. By comparing the differences in change among types, the driving forces causing these changes can be explored [24,25].

The calculation formula for the single land use dynamic index is:

$$K = \frac{U_b - U_a}{U_a} \times \frac{1}{T} \times 100\%,$$ (1)

Equation (1): K represents the dynamic index of a certain land use type in the study area during the study period; $U_a$ and $U_b$ are the areas of that land use type at the beginning and end of the study period, respectively; T is the length of the study period (in years).

**3.3.2. Comprehensive land use dynamic index.** The comprehensive land use dynamic index quantifies the overall speed and intensity of change for all land use types within the study area over a specific period [26,27].

The calculation formula for the comprehensive land use dynamic Index is:

$$M = \left\{ \sum_{ij}^{n} \left( \frac{\Delta S_{i-j}}{S_i} \right) \right\} \times \frac{1}{T} \times 100\%,$$ (2)

Equation (2): M is the comprehensive land use dynamic index within the region during the study period; $S_i$ is the area of land use type i at the beginning of the study period; $\Delta S_{i-j}$ represents the total area of land use type i converted to land use type j during the study period; n represents the number of land use types; T is the length of the study period (in years). To interpret the intensity of land use change, this study adopts the classification standards from previous research [28]. Referencing previous research, the comprehensive land use dynamic index is classified as follows: > 5% as a drastic change type; 0.30%−1.0% as a rapid change type; 0.10%−0.30% as a slow change type; < 0.10% as an extremely slow change type.

**3.3.3. Land use transition matrix.** The land use transition matrix quantitatively describes the conversion trends among various land use types, detailing the magnitudes and directions of transfers between specific types over a period [29–31].

The formula represents the land use transition matrix:

$$S_{ij} = \begin{vmatrix} S_{11} & \cdots & S_{1n} \\ \vdots & \vdots & \vdots \\ S_{n1} & \cdots & S_{nn} \end{vmatrix},$$ (3)

Equation (3): where S represents the area; n is the number of land use types; i and j represent the land use types at the beginning (row) and end (column) of the study period, respectively; $S_{ij}$ represents the area transferred from land use type i at the beginning of the period to land use type j at the end of the period.

## 3.4. Ecological sensitivity assessment

**3.4.1. Constructing the evaluation system and determining weights.** Based on previous research findings [32–34,41–46] and a full consideration of the regional characteristics of the Qingdao West Coast New Area, this study constructed an ecological sensitivity system comprising 7 influencing factors across three main aspects: natural environment, land cover, and accessibility (Table 2).

**Table 2. Construction of ecological sensitivity evaluation index system and indicator weights for the study area.**

| Target Layer | Criterion Layer | Indicator Layer | Weight |
|---|---|---|---|
| Ecological Sensitivity Evaluation Index System | Natural Environment | Elevation | 0.0535 |
| | | Slope | 0.0329 |
| | | Aspect | 0.0194 |
| | Land Cover | NDVI | 0.2335 |
| | | Land Use Type | 0.3678 |
| | Accessibility | Distance to Water Systems | 0.1434 |
| | | Distance to Roads | 0.0896 |

To quantify the relative importance of factors affecting regional ecological sensitivity, this study employed the Analytic Hierarchy Process (AHP) to determine indicator weights [35–37]. AHP structures complex problems hierarchically and quantifies judgments through pairwise comparisons.

To integrate expert knowledge and mitigate subjective bias, AHP was combined with the Delphi Method. Multiple experts from diverse fields (ecology, geography, urban and rural planning, etc.) performed pairwise indicator comparisons using Saaty's 1–9 scale [36,38] to construct judgment matrices. A comprehensive judgment matrix was derived using the geometric mean of corresponding elements from these expert matrices. Indicator weights were then calculated from this comprehensive matrix using Yaahp 10.3 software.

$$CI = \frac{\lambda_{max} - n}{n - 1}.$$ (4)

$$CR = \frac{CI}{RI},$$ (5)

Equation (4) and (5): CI represents the Consistency Index, $\lambda_{max}$ is the maximum eigenvalue, RI is the Random Index (a fixed value obtainable from standard random consistency index tables), and N is the number of selected ecological sensitivity factors. If the Consistency Ratio (CR = CI/RI) < 0.1, it indicates that the judgment matrix passes the consistency test, and the allocation of ecological factor weights is reasonable.

We performed a consistency test on the final composite judgment matrix. The results showed CR = 0.0081 < 0.1 (Table 3), passing the test and indicating that the weight allocation is reasonable.

As determined by the AHP method (Table 2), Land Use Type (0.3678) and NDVI (0.2335) were the most influential factors in the ecological sensitivity assessment.

**3.4.2. Ecological sensitivity factor grading criteria.** Our grading criteria were established by following the relevant regulations of the "National Ecological Function Zoning" and the "Guidelines for Delineating Ecological Protection Red Lines" [39,40], referencing prior research on evaluation indicators and classification standards [31–34,41–46], and combining the Natural Breaks method with expert experience. Based on this framework, the ecological sensitivity of the West Coast New Area was divided into five levels: non-sensitive, low sensitivity, moderate sensitivity, high sensitivity, and extremely high sensitivity, which were assigned corresponding values (Table 4).

**Table 3. Summary of Consistency Check Results.**

| $\lambda_{max}$ | CI | RI | CR |
|---|---|---|---|
| 7.0639 | 0.0107 | 1.32 | 0.0081 |

**Table 4. Ecological Sensitivity Factor Grading Criteria.**

| Ecological Factor | Non-sensitive | Low Sensitivity | Moderate Sensitivity | High Sensitivity | Extremely High Sensitivity |
|---|---|---|---|---|---|
| Elevation/ m | <50 | 50~150 | 150~300 | 300~450 | >450 |
| Slope/ (°) | <5 | 5~15 | 15~25 | 25~35 | >35 |
| Aspect | Flat ground, South | Southeast, Southwest | East, West | Northeast, Northwest | North |
| NDVI | <0.2 | 0.2-0.4 | 0.4-0.6 | 0.6-0.8 | >0.8 |
| Land Cover | Construction land | Unused land | Cultivated land | Grassland, Water body | Forest land |
| Distance to Roads/ m | >1500 | 1000-1500 | 500-1000 | 200-500 | <200 |
| Distance to Water Systems/ m | >1000 | 500-1000 | 300-500 | 100-300 | <100 |
| Assigned Value | 1 | 3 | 5 | 7 | 9 |

**1)** Topographic Factors: Ecological sensitivity increases with higher elevations and steeper slopes, primarily due to greater soil erosion risks, challenges in vegetation recovery, and potential biodiversity significance. North-facing slopes received the highest sensitivity rating because of distinct microclimatic conditions, such as reduced solar radiation and potentially different moisture availability. **2)** Land cover type directly indicates the complexity and stability of the ecosystem. Forest ecosystems, with their complex structure, high biodiversity, and critical roles in soil and water conservation, are highly susceptible to irreversible damage from disturbances, thus warranting the highest sensitivity rating. This is followed by grasslands and water bodies. Conversely, built-up land, being an already highly disturbed and artificial system, is deemed non-sensitive to further ecological disruption. **3)** Accessibility Factors: These factors gauge the intensity of potential human disturbance. Closer proximity to roads and water systems implies increased exposure to impacts like traffic, pollution, and development, rendering ecosystems more vulnerable and thus assigned higher sensitivity.

Taken together, these classification standards incorporate both the inherent natural characteristics of the region and the influence of human-induced stressors on environmental stability

## 3.5. Patch-generating Land Use Simulation (PLUS) model

Land use change simulation is crucial for understanding and predicting regional development trends and for evaluating policy impacts. While previous studies have widely employed classic models like the Future Land-Use Simulation (FLUS) and Cellular Automata (CA) models [28,47], they often have limitations in capturing complex change dynamics.

To more accurately explore the underlying drivers of land use change and improve simulation accuracy, this study selected the PLUS (Patch-generating Land Use Simulation) model, a new-generation model developed by Liang Xun et al. The core innovation of the PLUS model is its coupling of two key modules: the Land Expansion Analysis Strategy (LEAS) and a CA model based on a Random Seeds and Patch Generation mechanism (CARS) [48,49].

The LEAS module, which integrates a Random Forest algorithm, offers a significant advantage by automatically uncovering the complex, non-linear relationships between land expansion and its various drivers, a capability that enhances causal inference compared to traditional regression models. Meanwhile, the CARS module excels at simulating dynamic landscape changes. By using a random seed generation and threshold-decreasing mechanism, this module can realistically simulate the spontaneous generation and growth of land patches based on their development probabilities [49]. This powerful capability allows the model to better explain the change mechanisms of different land use types, leading to improved simulation results and higher accuracy.

As demonstrated in multiple studies, the PLUS model provides higher simulation accuracy and a better explanation of land use change mechanisms compared to models like FLUS and CA-Markov [50], making it the ideal tool for our analysis.

**3.5.1. Model accuracy validation.** To verify the accuracy of the model simulation, the Kappa coefficient and the Fig of Merit (FOM) were selected as evaluation metrics. Generally, a Kappa coefficient greater than 0.75 indicates that the simulation results are relatively accurate, while a larger FOM coefficient indicates higher simulation accuracy [49].

**3.5.2. Setting up future multi-scenario modes.** Three different scenarios were set up to simulate the spatial layout of land use in the West Coast New Area for the year 2030. A key constraint applied to all scenarios was the protection of major water bodies (e.g., rivers, lakes, and reservoirs), which are strictly protected under government policy and are unlikely to be converted [51–53].

1) Natural Development Scenario (ND):

This scenario projects future land use demand using the Markov-chain method, assuming a continuation of trends observed from 2010–2020. The transition matrix between land use types was determined, and the neighborhood weights for the conversion of each land use type were calculated (Equation (6)). This scenario serves as the baseline for the other two scenarios [49].

$$W_i = \frac{T_i - T_{min}}{T_{max} - T_{min}}$$

(6)

Equation (6): $W_i$ is the neighborhood weight for land use type i, $T_i$ is the expansion area of land use type i, and $T_{min}$ and $T_{max}$ are the minimum and maximum expansion areas among all land use types.

2) Urban Development Scenario (UD):

This scenario simulates a future where economic development is prioritized, leading to rapid urbanization and industrialization. Building upon the ND scenario, the transition matrix is adjusted to favor urban growth. The probability of construction land converting to other land types is reduced by 40%. In contrast, the probabilities of cultivated land, forest land, grassland, water bodies, and unused land converting to construction land increase by 40%, 10%, 20%, 10%, and 50%, respectively [51–57].

3) Ecological Protection Scenario (EP):

This scenario is designed to align with the goals of key policy documents, including the "Planning for the National Ecological Civilization Construction Demonstration Zone in Qingdao West Coast New Area (2020-2030)", the "Territorial Spatial Zoning Plan for Qingdao West Coast New Area (2021-2030)", and "The 14th Five-Year Plan for National Economic and Social Development and the Long-Range Objectives Through the Year 2030 for Qingdao City" [58,59].

In this scenario, urban development is moderated to ensure ecosystem security. The transition probabilities from forest land and grassland to construction land are reduced by 50%, the transition probability from cultivated land to construction land is reduced by 30%, and the transition probability from construction land to forest land is increased by 10% [51–57]. As an additional constraint, the areas identified as 'extremely high sensitivity' are protected from conversion.

## 4. Results analysis and discussion

### 4.1. Spatio-temporal evolution of land use

**4.1.1. Single land use dynamic index.** Cultivated land showed a significant trend of continuous and accelerating loss (Fig 4, Table 5). In the three periods of 1990–2000, 2000–2010, and 2010–2020, the net decrease in cultivated land area was 6951.69 ha, 8875.89 ha, and 11378.16 ha, respectively. This indicates that the cultivated land area was continuously shrinking, with the reduction in each subsequent period being more significant than in the previous one. Correspondingly, the single land use dynamic index for cultivated land remained negative throughout the three periods, and its absolute value showed an increasing trend (i.e., the negative change intensified). The specific value decreased from −0.44% in

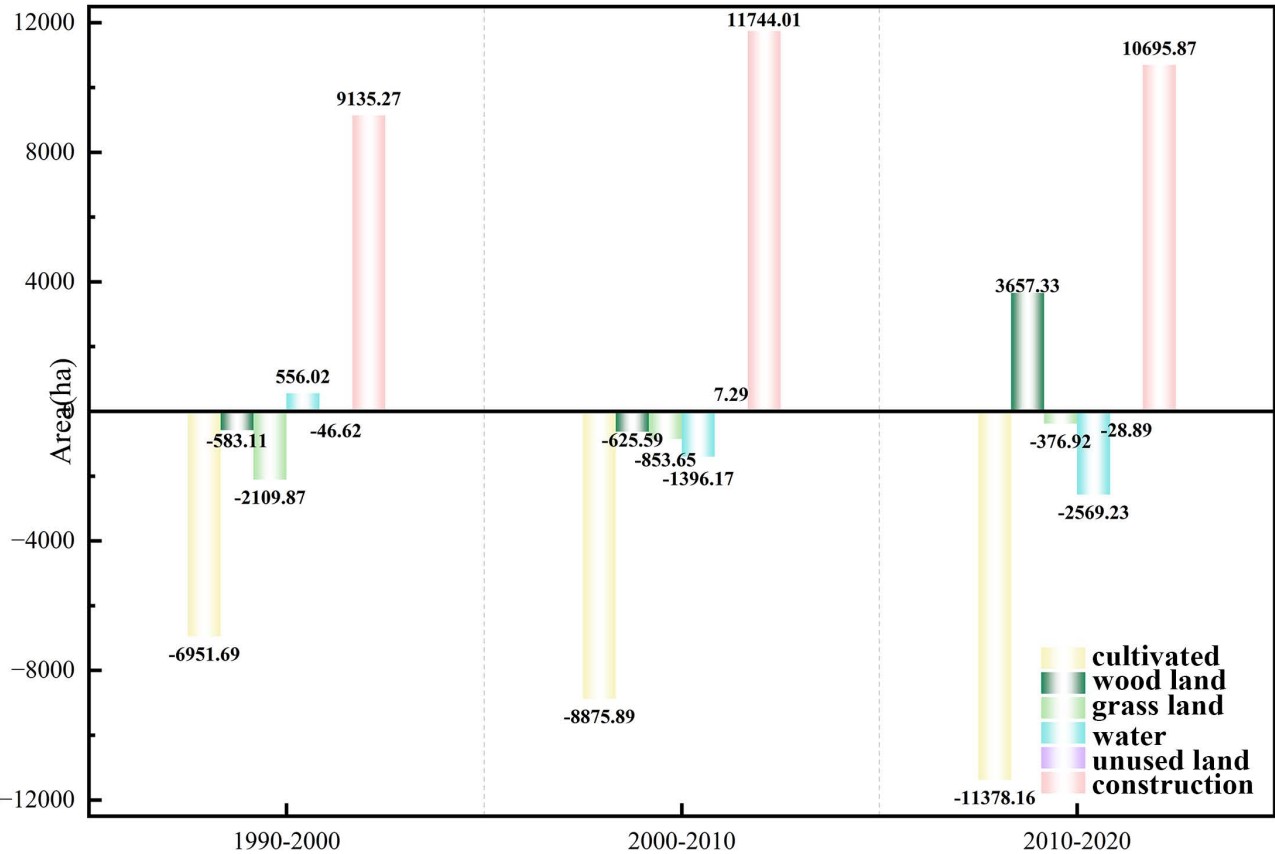

**Fig 4. Changes in Land Use Area (1990–2020).**

**Table 5. Single land use dynamic index in the West Coast New Area.**

| Time | 1990-2000 | 2000-2010 | 2010-2020 |
|---|---|---|---|
| Land use type | Single land use dynamic index | | |
| Cultivated Land | −0.44% | −0.59% | −0.80% |
| Forest land | −0.53% | −0.60% | 3.72% |
| Grass Land | −5.50% | −4.94% | −4.30% |
| Water Body | 0.53% | −1.27% | −2.67% |
| Unused land | −3.67% | 0.91% | −3.30% |
| Construction land | 3.59% | 3.39% | 2.31% |

1990–2000 to −0.59% in 2000–2010 and further decreased to −0.80% in 2010–2020. This intuitively reflects that the average annual rate of decrease in cultivated land area progressively accelerated during the study period.

Meanwhile, construction land showed a trend of large-scale expansion. In 1990–2000, 2000–2010, and 2010–2020, the net increase in construction land area was 9135.27 ha, 11744.01 ha, and 10695.87 ha, respectively. This demonstrates that construction land underwent significant growth throughout the study period, with the most rapid expansion occurring between 2000 and 2010.

In summary, the continuous rapid reduction of cultivated land and the significant expansion of construction land are the most critical characteristics of land use change in the West Coast New Area over the past thirty years. These two types exhibit a clear inverse relationship (one decreasing as the other increases), directly reflecting the profound impact of the region's rapid urbanization and industrialization on the land use structure.

**4.1.2. Comprehensive land use dynamic index.** The results (Fig 5) indicate that the study area consistently exceeded >5% during the different analysis periods, uniformly falling into the "drastic change type" category. More notably, the value of this dynamic index indicator also exhibited a trend of increasing in each subsequent period. This clearly reveals that the study area not only experienced high-intensity land use change over the past three decades, but that the rate of this change has also been continuously accelerating.

Against this background of drastic overall change, conversions between different land use types were frequent and complex, with the mutual conversion between cultivated land and construction land constituting the primary flow of change. According to the statistics of land area by type during the study period (Table 6), the cultivated land area significantly shrank from 157855.32 ha in 1990 to 130649.58 ha in 2020, a net decrease of over 27000 ha. Meanwhile, construction land experienced explosive growth, expanding from 25479.54 ha in 1990 to 57054.69 ha in 2020, a net increase of over 31500 ha.

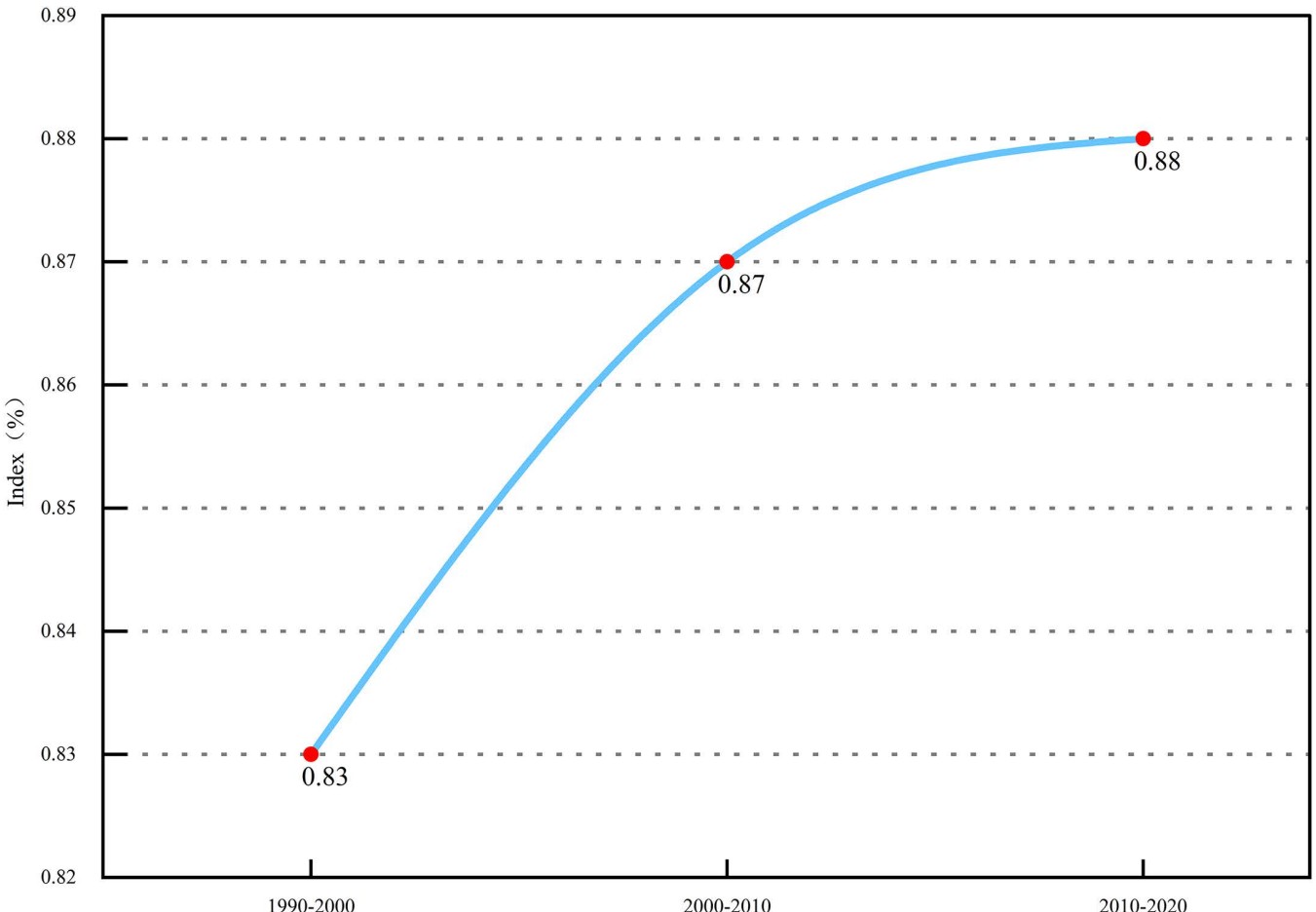

**Fig 5. Comprehensive Land Use Dynamic Index.**

**Table 6. Land use area of each period in the study area from 1990 to 2020 (ha).**

| Time | Cultivated land | Forest Land | Grass Land | Water Body | Unused Land | Construction Land |
|------|-----------------|-------------|------------|------------|-------------|-------------------|
| 1990 | 157855.32 | 11034.54 | 3839.13 | 10465.38 | 126.99 | 25479.54 |
| 2000 | 150903.63 | 10451.43 | 1729.26 | 11021.40 | 80.37 | 34614.81 |
| 2010 | 142027.74 | 9825.84 | 875.61 | 9625.23 | 87.66 | 46358.82 |
| 2020 | 130649.58 | 13483.17 | 498.69 | 7056.00 | 58.77 | 57054.69 |

This evolution of the land use structure, primarily characterized by the continuous decrease in cultivated land and the drastic expansion of construction land, is inextricably linked to the economic takeoff and social transformation experienced by the Qingdao West Coast New Area during this period. The period from 1990 to 2020 represented the "golden three decades" of economic and social development for the region. During this time, the West Coast New Area transitioned from the original county-level administrative units of Jiaonan City and Huangdao District to a national-level new area. This elevation in strategic status significantly promoted favorable policies, capital accumulation, and industrial development. The regional gross economic product repeatedly reached new highs, and urbanization advanced unprecedentedly. By 2020, the New Area's Gross Domestic Product (GDP) had reached 372.168 billion RMB, accounting for nearly one-third of the total GDP of Qingdao City. Its comprehensive economic strength ranked third among China's 19 national-level new areas, surpassed only by Shanghai Pudong New Area and Tianjin Binhai New Area.

Rapid industrialization and urbanization inevitably required an immense demand for industrial, commercial, residential, and infrastructure land, thereby directly driving the large-scale expansion of construction land, which often occurred at the expense of encroaching upon surrounding prime cultivated land. Therefore, the rapid growth of the regional macroeconomy and significant adjustments in development strategy are the fundamental driving forces behind the drastic changes in the study area's land use patterns, particularly the substantial loss of cultivated land and the rapid increase in construction land.

**4.1.3. Land use transition matrix.** Analysis of the land use transition matrix (Table 7, Fig 6) for 1990–2000 reveals that land use change in the study area began to show noticeable activity. Cultivated land was the primary conversion source, with 10033.33 ha transferred, mainly flowing to construction land, water bodies, and forestland. At the same time, construction land was the primary expanding land type, with a net increase of 9135.27 ha during this decade. Its expansion mainly originated from the conversion of cultivated land, grassland, and forestland. This pattern reflects that the study area entered the initial stage of rapid urbanization in the 1990s, with urban construction exerting pressure on surrounding farmland and some ecological spaces.

Analysis of the land use transition matrix (Table 8, Fig 6) for 2000–2010 reveals that the intensity of land use change in the study area significantly increased. The scale of cultivated land conversion expanded sharply, transferring as much

**Table 7. Land use type transfer matrix in the Study Area from 1990 to 2000 (ha).**

| Item | Cultivated Land | Forest land | Grass Land | Water Body | Unused land | Construction land | 1990 in Total |
|------|-----------------|-------------|------------|------------|-------------|-------------------|---------------|
| Cultivated Land | 147822.12 | 646.38 | 247.32 | 907.56 | 0.09 | 8231.85 | 157855.32 |
| Forest land | 2143.26 | 8768.07 | 66.87 | 0.54 | 0.00 | 55.80 | 11034.54 |
| Grass Land | 886.41 | 1036.62 | 1412.28 | 28.44 | 19.62 | 455.76 | 3839.13 |
| Water Body | 49.68 | 0.09 | 0.63 | 8987.76 | 4.86 | 1422.36 | 10465.38 |
| Unused land | 0.81 | 0.00 | 2.16 | 8.46 | 55.80 | 59.76 | 126.99 |
| Construction land | 1.35 | 0.27 | 0.00 | 1088.64 | 0.00 | 24389.28 | 25479.54 |
| 2000 in Total | 150903.63 | 10451.43 | 1729.26 | 11021.40 | 80.37 | 34614.81 | 208800.90 |

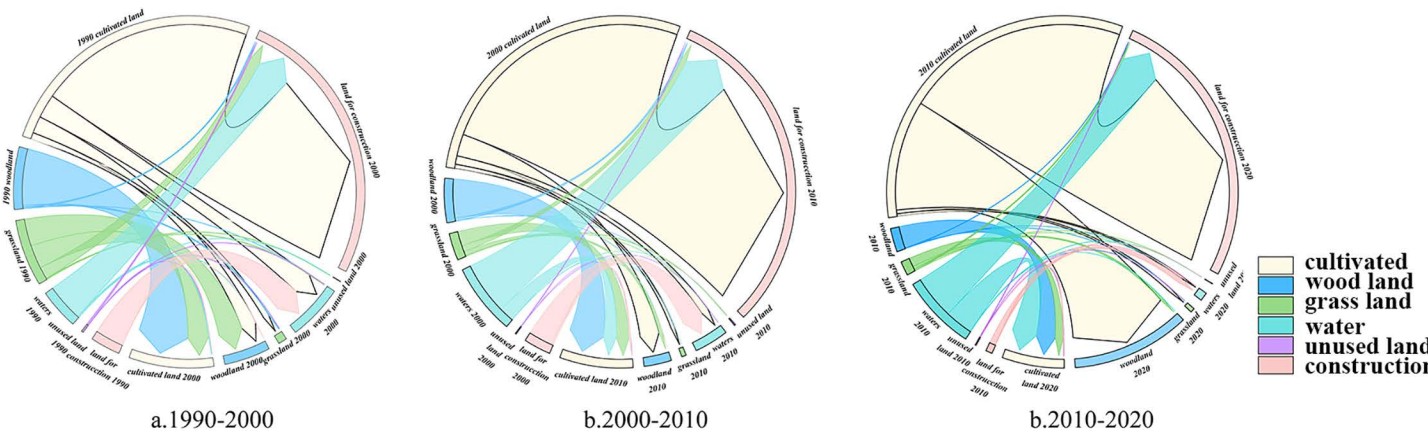

Fig 6. Land Use Transition Chord Diagram in the Study Area from 1990 to 2020.

Table 8. Land use type transfer matrix in the Study Area from 2000 to 2010(ha).

| Item | Cultivated Land | Forest land | Grass Land | Water Body | Unused land | Construction land | 1990 in Total |
|---|---|---|---|---|---|---|---|
| Cultivated Land | 139244.58 | 870.84 | 165.96 | 337.68 | 1.35 | 10283.22 | 150903.63 |
| Forest land | 1542.24 | 8784.63 | 5.40 | 0.09 | 0.00 | 119.07 | 10451.43 |
| Grass Land | 580.05 | 170.01 | 703.35 | 5.22 | 30.60 | 240.03 | 1729.26 |
| Water Body | 585.54 | 0.27 | 0.90 | 8326.17 | 9.99 | 2098.53 | 11021.40 |
| Unused land | 0.45 | 0.00 | 0.00 | 3.69 | 42.30 | 33.93 | 80.37 |
| Construction land | 74.88 | 0.09 | 0.00 | 952.38 | 3.42 | 33584.04 | 34614.81 |
| 2000 in Total | 142027.74 | 9825.84 | 875.61 | 9625.23 | 87.66 | 46358.82 | 208800.90 |

as 11,659.05 ha of cultivated land primarily to construction land, water bodies, and forestland. Construction land saw the largest growth, increasing by 11,744.01 ha from 2000 to 2010. Correspondingly, the expansion of construction land also peaked during this period with this net increase. It is worth noting that, in addition to the continued large-scale conversion of cultivated land, grassland, and forestland, the proportion of water bodies converted to construction land also increased significantly. This indicates that as the urbanization process accelerated, the encroachment on traditional farmland and forest/grassland intensified, and it even began to encroach upon some water body spaces, making land use conflicts more prominent.

Analysis of the land use transition matrix (Table 9, Fig 6) for 2010–2020 reveals that the overall pattern of land use change in the study area continued, but with shifts in intensity and structure. Cultivated land remained the most significant source land type, with a transferred area of 13,774.14 ha, mainly converting to construction land and forestland. Construction land continued its expansion trend, but its expansion rate slowed down, with a net increase of 10,695.87 ha over the decade. Cultivated land, grassland, and water bodies provided the main areas for this expansion.

Significantly different from the previous two stages, the area of forestland showed a marked increase (net increase of 3,657.33 ha). This is closely related to the increasing emphasis on ecological and environment protection at the national and local levels, the implementation of ecological construction projects such as "Grain for Green," and possibly existing urban development boundary control policies. The relative slowdown in the growth rate of construction land might also corroborate a development model shifting towards greater emphasis on quality and sustainability.

**Table 9. Land use type transfer matrix in the Study Area from 2010 to 2020(ha).**

| Item | Cultivated Land | Forest land | Grass Land | Water Body | Unused land | Construction land | 1990 in Total |
|------|-----------------|-------------|------------|------------|-------------|-------------------|---------------|
| Cultivated Land | 128253.60 | 4441.41 | 151.47 | 102.24 | 0.18 | 9078.84 | 142027.74 |
| Forest land | 873.90 | 8917.11 | 5.04 | 0.09 | 0.00 | 29.70 | 9825.84 |
| Grass Land | 354.33 | 119.25 | 333.27 | 0.18 | 0.90 | 67.68 | 875.61 |
| Water Body | 1129.68 | 5.40 | 0.18 | 6726.60 | 12.69 | 1750.68 | 9625.23 |
| Unused land | 8.73 | 0.00 | 8.73 | 0.18 | 44.82 | 25.20 | 87.66 |
| Construction land | 29.34 | 0.00 | 0.00 | 226.71 | 0.18 | 46102.59 | 46358.82 |
| 2000 in Total | 130649.58 | 13483.17 | 498.69 | 7056.00 | 58.77 | 57054.69 | 208800.90 |

Overall, the land use transitions in the study area over the past thirty years exhibit a continuous evolution dominated by urbanization, persistently converting cultivated land (the primary source) to support the rapid expansion of construction land. However, the intensity and internal structure of the changes were not constant. The period of 2000–2010 was the most drastic change phase, during which construction land expansion peaked and began to affect water bodies. Conversely, from 2010–2020, although the core trend remained unchanged, the expansion rate of construction land tended to slow down. Simultaneously, the restorative growth of ecological land (forestland) became a new highlight, reflecting the dynamic adjustment of the interactive relationship between economic development and ecological protection.

## 4.2. Ecological sensitivity assessment

**4.2.1. Single factor ecological sensitivity assessment.** Given the inherent stability of topographic (elevation, slope, aspect) and hydrological factors within the region, data from a single reference year provided an adequate basis for their assessment in the ecological sensitivity analysis. Therefore, employing ArcGIS 10.8 software and the single-factor grading standards from Table 4, we evaluated the ecological sensitivity based on the obtained data for elevation, slope, aspect, vegetation cover, land cover, road distance, and water systems. The resulting classification maps for each factor's sensitivity are presented in Fig 7.

**4.2.2. Comprehensive ecological sensitivity assessment.** The comprehensive ecological sensitivity of the study area was computed using the ArcGIS 10.8 Raster Calculator via a weighted overlay method, integrating single-factor weights and their spatial distribution data. Based on the criteria in Table 10, the results were categorized into five sensitivity levels: 1 (Insensitive), 2 (Low Sensitivity), 3 (Moderate Sensitivity), 4 (High Sensitivity), and 5 (Extreme Sensitivity). The spatial distribution of these levels is presented in Fig 8.

Fig 8 shows that the study area's ecological sensitivity exhibits evident spatial differentiation characteristics regarding spatial distribution.

Low Sensitivity Areas (Non-sensitive and Low Sensitivity Areas): These cluster mainly in the study area's northeastern, central, and southeastern coastal zones, corresponding to the core ranges of the three major functional clusters: Huangdao, Jiaonan, and Dongjiakou. These areas are characterized by relatively flat terrain, low elevation, and high concentrations of urban and industrial activity. Due to the high degree of development, the land use structure has undergone profound changes, resulting in strong resistance to disturbance and low ecological sensitivity.

Moderate Sensitivity Areas: In terms of spatial distribution, these areas function as critical transition zones between the urbanized plains and the protected mountain regions. Ecologically, they are predominantly composed of cultivated land and peri-urban landscapes, which are subject to both agricultural activities and pressures from urban expansion. Their interspersed distribution pattern highlights their role as the primary frontier where land use conflicts between development and conservation occur.

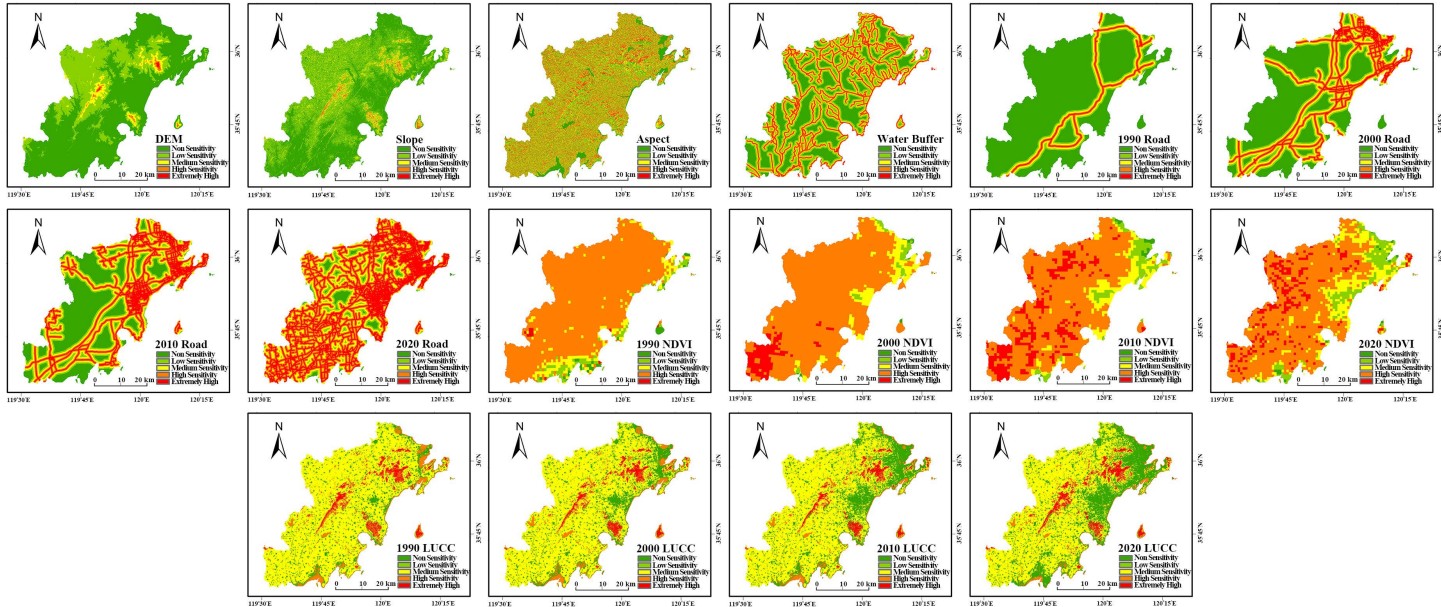

**Fig 7. Spatial distribution of single-factor ecological sensitivity levels in the study area.**

**Table 10. Comprehensive Ecological Sensitivity Grading Criteria of the study area.**

| Evaluation Grade | Numerical Assignment | Comprehensive Grading Criteria | Grade Description |
|---|---|---|---|
| I | 1 | <3 | non-sensitive |
| II | 2 | 3~4 | low sensitivity |
| III | 3 | 4~5 | moderate sensitivity |
| IV | 4 | 5~6 | high sensitivity |
| V | 5 | >6 | extremely high sensitive |

High Sensitivity Areas (High and Extremely High Sensitivity Areas): These cluster mainly around mountain ranges such as Xiaozhu Mountain, Cangma-Tiejue Mountain, and Dazhu Mountain. These areas typically have higher elevations, complex terrain, and ecosystems that are relatively pristine or fragile, thus exhibiting weak resistance to external disturbances. This results in their high ecological sensitivity. It is a key finding that these highly sensitive zones show a strong spatial correlation with designated land uses such as ecological protection zones, water source conservation areas, and important tourism resource sites, including numerous national parks and scenic spots. Human activity is relatively limited in these zones, which contributes to the preservation of their sensitive ecological characteristics.

Temporal Changes in Sensitivity Patterns (1990–2020):

Statistics compiled using ArcGIS 10.8 (Table 11) reveal that the ecological sensitivity pattern of the study area underwent significant changes over the past thirty years.

Low-Sensitivity Areas: The area of non-sensitive zones continuously expanded, increasing from 13,302.88 ha (6.37%) in 1990–20,978.76 ha (10.05%) in 2020, directly reflecting ongoing urbanization. Conversely, the area of low-sensitivity zones decreased from 40,803.17 ha (19.54%) to 32,189.79 ha (15.42%).

Moderate-Sensitivity Zones: This zone experienced the most drastic change, shrinking sharply from 117,022.44 ha (56.04%) in 1990–79,416.97 ha (38.03%) in 2020.

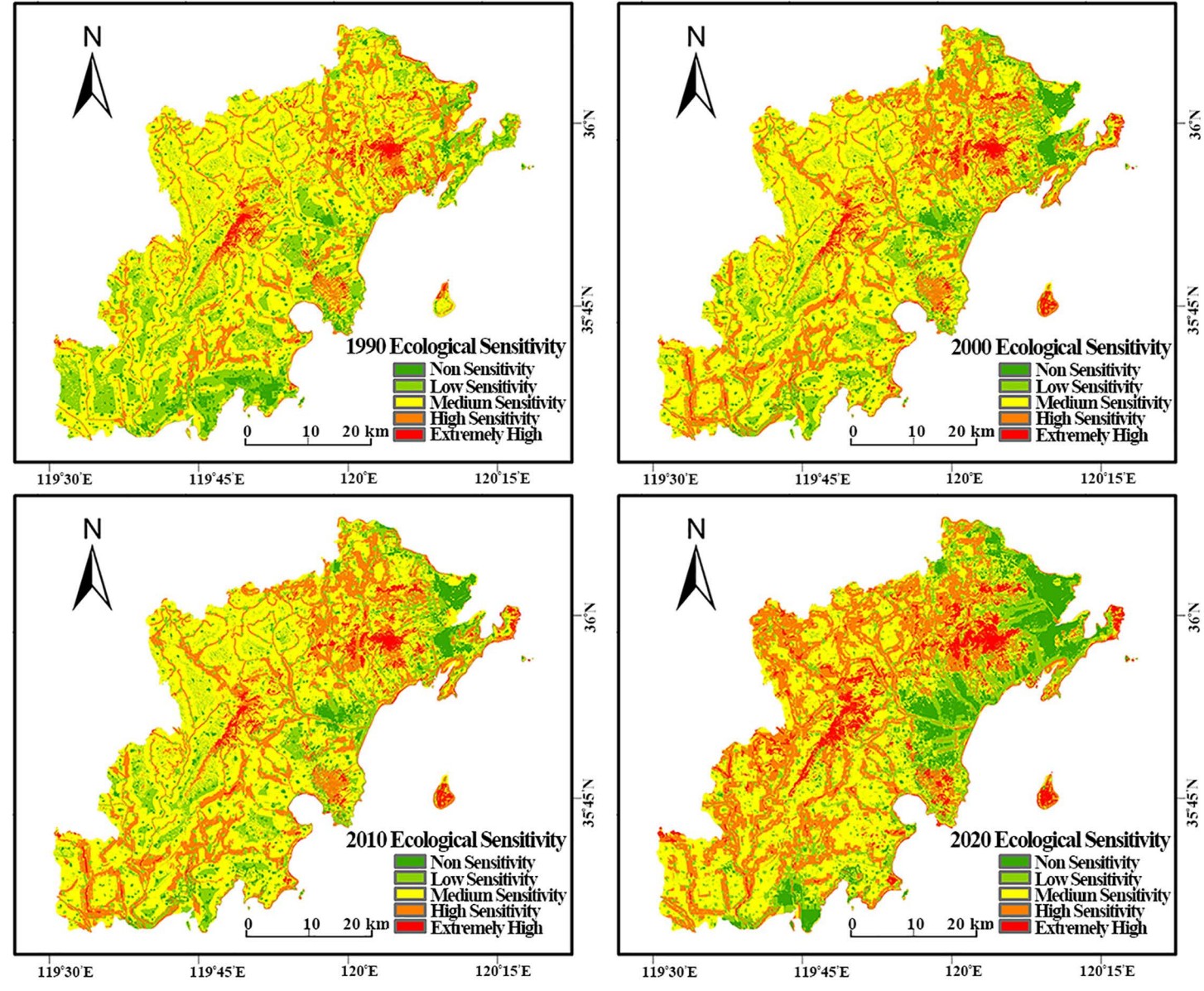

**Fig 8. Spatial distribution of integrated ecological sensitivity levels in the study area from 1990 to 2020.**

High-Sensitivity Zones: Corresponding to the decrease in moderately sensitive zones, high and extremely high sensitivity zones showed significant growth. The total area proportion of these zones increased substantially from approximately 18% in 1990 to 36.50% in 2020.

The Spatial Polarization of Ecological Sensitivity:

This study's core finding is the "spatial polarization of the ecological sensitivity pattern," a seemingly contradictory phenomenon where both 'non-sensitive' and 'highly sensitive' areas increased concurrently. This pattern is not a methodological artifact but rather a logical consequence of China's unique spatial planning paradigm, where 'development' and 'conservation' are spatially decoupled. This finding aligns with recent studies on coastal urbanization in China [1],

**Table 11. Area of each sensitive area in different years and the proportion of the study area from1990 to 2020(ha).**

| Time | 1990 | | 2000 | | 2010 | | 2020 | |
|---|---|---|---|---|---|---|---|---|
| | Area | % | Area | % | Area | % | Area | % |
| non-sensitive | 13302.88 | 6.37% | 12538.98 | 6.01% | 18543.86 | 8.88% | 20978.76 | 10.05% |
| low sensitivity | 40803.17 | 19.54% | 31018.43 | 14.86% | 32600.96 | 15.61% | 32189.79 | 15.42% |
| moderate sensitivity | 117022.44 | 56.04% | 110408.45 | 52.88% | 94928.86 | 45.46% | 79416.97 | 38.03% |
| high sensitivity | 33471.97 | 16.03% | 48726.39 | 23.34% | 55384.89 | 26.53% | 62823.29 | 30.09% |
| extremely high sensitive | 4200.44 | 2.01% | 6108.64 | 2.93% | 7342.33 | 3.52% | 13392.08 | 6.41% |
| Total | 208800.90 | 1 | 208800.90 | 1 | 208800.90 | 1 | 208800.90 | 1 |

yet offers a more distinct characterization of the "polarization" effect compared to the general fragmentation observed in inland regions [3]. It is driven by two powerful, parallel, and spatially explicit forces:

Driver One: The National-Level New Area Development Strategy, Concentrated in Coastal Plains.

This policy-driven, high-intensity urbanization process is spatially concentrated in the northeastern, central, and southeastern coastal zones, which are characterized by flat terrain, high concentrations of economic activity, and the most intensive urban construction—namely, the three core economic functional clusters of Huangdao, Jiaonan, and Dongjiakou. The rapid urban and industrial expansion inevitably requires vast amounts of land, primarily sourced from the encroachment upon surrounding agricultural lands and transitional zones of moderate ecological sensitivity. This directly leads to a significant expansion of 'non-sensitive' areas, which represent high levels of anthropogenic disturbance, within these specific zones.

Driver Two: The Mandatory Ecological Conservation Policy, Focused on Inland Mountainous Regions.

Concurrently, mandatory conservation policies, epitomized by the "ecological protection red line," have precisely targeted the region's most critical ecological function areas: the three core inland mountain systems of Xiaozhu Mountain, Dazhu Mountain, and Cangma -Tiejue Mountain. These policies have provided strict legal protection for these areas and have promoted a series of ecological restoration projects, such as afforestation and mountain habitat rehabilitation. According to this study's evaluation framework, high vegetation cover and forest land are the core factors assigned the highest weights for 'high sensitivity'; therefore, these successful ecological restoration efforts are accurately identified by our model as a significant increase in the extent of 'highly sensitive' and 'extremely highly sensitive' zones.

Therefore, this 'polarization' pattern authentically reflects the unique spatial reality of China's rapid development, where 'high-intensity construction' and 'mandatory conservation' operate as parallel strategies in functionally zoned spaces. The direct trade-off is the rapid transformation and squeezing of the vast 'moderately sensitive areas' that act as intermediate buffer zones, ultimately resulting in a landscape where development and conservation zones are highly differentiated and spatially explicit. This serves as strong evidence for the effectiveness of our assessment framework, as it successfully captures this complex and policy-driven regional evolution, a capability often limited in conventional static assessments [4].

### 4.3. Future scenario simulation and assessment (2030)

**4.3.1. Validation of PLUS model accuracy.** To ensure the applicability and predictive reliability of the PLUS model in the West Coast New Area, this study first conducted an accuracy validation. The procedure was as follows: historical land use data from 2000 and 2010 were used as inputs to simulate the land use status for 2020. The simulated 2020 map was then compared spatially and quantitatively against the actual 2020 land use data to rigorously evaluate the model's simulation accuracy.

Initially, based on data availability and relevance to land use change, 11 driving factors were selected for the simulation, covering aspects of topography (elevation, slope, aspect), location (distance to water bodies, distance to roads), climate,

and vegetation (annual average precipitation, NDVI, annual average temperature), and socio-economics (population density, GDP, nighttime lights) (Fig 9). Key model parameters, such as neighborhood weights (Table 12) and land use demand for 2020 (Table 13), were determined using the 2000–2010 historical data and the integrated Markov Chain module.

The accuracy validation confirmed the model's robustness through a comprehensive framework. The comparison yielded a Kappa coefficient of 0.782, an Overall Accuracy (OA) of 0.886, and a Fig of Merit (FoM) of 0.0834 (Fig 10). The high OA value confirms the model's global reliability, while the FoM index specifically validates the model's capability in capturing dynamic spatial changes, distinguishing it from mere quantity agreement. These metrics signify a high degree of simulation accuracy and strong consistency in both quantitative allocation and spatial distribution, validating the model's suitability for simulating future land use scenarios. The successful validation of the PLUS model in this complex coastal setting underscores its utility for regional planning, supporting findings from similar applications in other rapidly urbanizing areas [48].

**4.3.2. Future land use scenarios (2030).** Using the validated PLUS model and 2010–2020 land use data, we simulated the spatial layout of land use for 2030 under three scenarios: Natural Development (ND), Urban Development (UD), and Ecological Protection (EP). The simulation process involved three main steps: (1) generating development probability maps for each land use type using the LEAS module and Random Forest algorithm (Fig 11); (2) calculating neighborhood weights based on the 2010–2020 transition matrix (Table 14); and (3) determining land use demands for each scenario (Table 15) and allocating them spatially using the CARS module (Fig 12).

Natural Development (ND) Scenario: This scenario projects a continuation of recent trends (2010–2020). The main characteristic is the continued reduction of cultivated land (projected decrease of 10,399.59 ha). Meanwhile, forestland is projected to increase by 2,910.51 ha, and construction land will continue to grow, indicating that development pressure on agricultural land and ecological restoration efforts coexist.

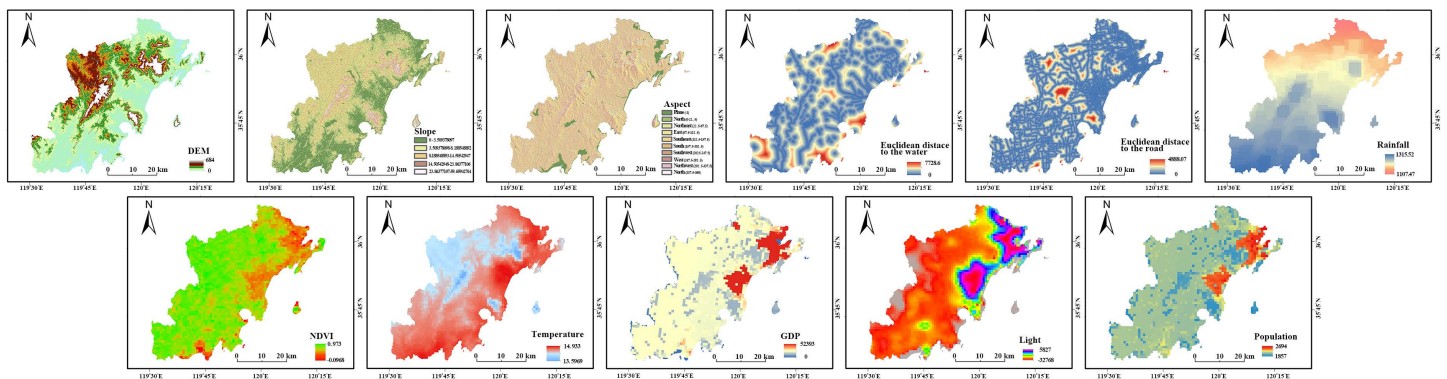

**Fig 9. Driving Factors.**

**Table 12. Neighborhood Weights.**

| Type | Cultivated land | Forest Land | Grass Land | Water Body | Unused Land | Construction Land |
|------|-----------------|-------------|------------|------------|-------------|-------------------|
| Weight | 0.756 | 0.0526 | 0.072 | 0.018 | 0.001 | 1 |

**Table 13. 2020 Actual vs Simulated Area (ha).**

| Type | Cultivated land | Forest Land | Grass Land | Water Body | Unused Land | Construction Land |
|------|-----------------|-------------|------------|------------|-------------|-------------------|
| 2020 Actual | 130649.58 | 13483.17 | 498.69 | 7056.00 | 58.77 | 57054.69 |
| 2020 Simulated | 133410.24 | 9164.88 | 518.22 | 8871.48 | 76.23 | 56759.85 |

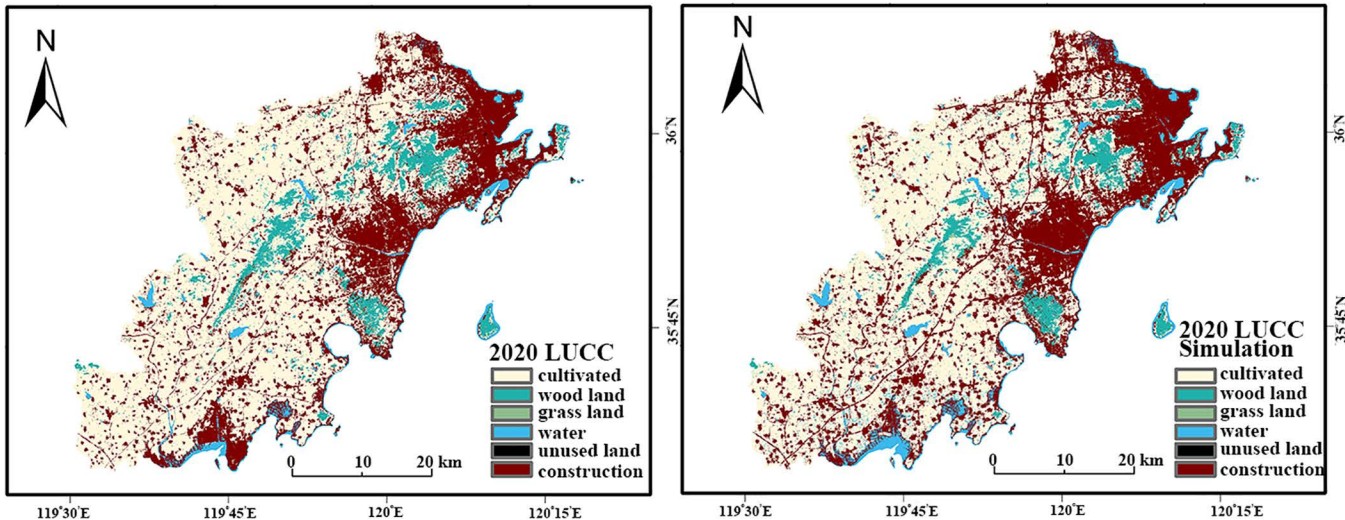

**Fig 10. Actual Land Use Data (2020) vs. Simulated Land Use Data (2020).**

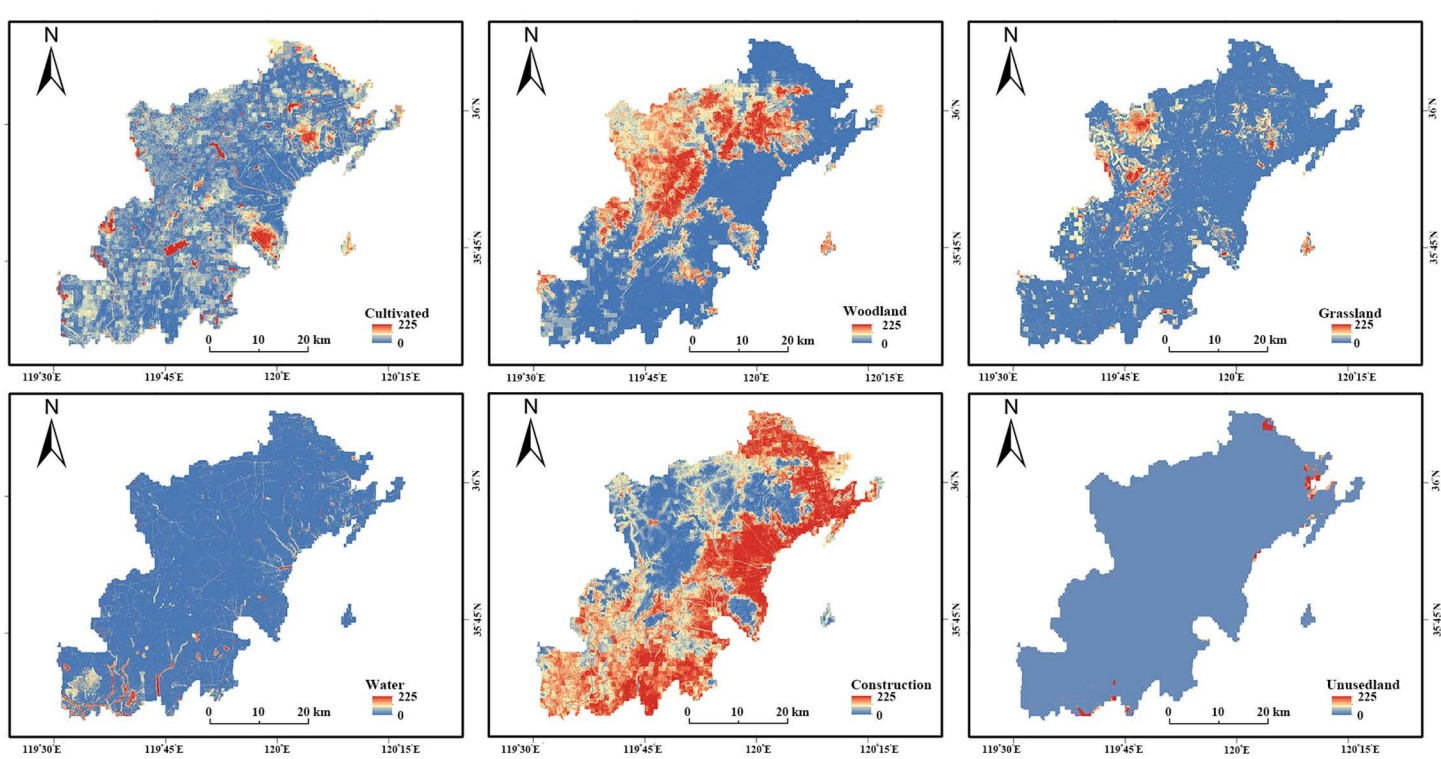

**Fig 11. Atlas of Development Probability for Different Land Use Types.**

Urban Development (UD) Scenario: Under this scenario, the expansion of construction land becomes the dominant trend, with a projected increase of 13,031.06 ha—the largest among the three scenarios. This growth places immense pressure on cultivated land, which is projected to decrease sharply by 13,659.86 ha. While this scenario may yield

**Table 14. Neighborhood Weights.**

| Type | Cultivated land | Forest Land | Grass Land | Water Body | Unused Land | Construction Land |
|------|-----------------|-------------|------------|------------|-------------|-------------------|
| Weight | 1 | 0.319 | 0.031 | 0.224 | 0.001 | 0.939 |

**Table 15. Prediction of Land Use Areas in 2030 Under Different Development Scenarios (ha).**

| Type | Cultivated land | Forest Land | Grass Land | Water Body | Unused Land | Construction Land |
|------|-----------------|-------------|------------|------------|-------------|-------------------|
| ND | 120249.99 | 16393.68 | 342.00 | 5304.51 | 40.23 | 66470.40 |
| UD | 116989.73 | 16277.08 | 333.81 | 5080.68 | 33.85 | 70085.75 |
| EP | 122677.38 | 16498.71 | 352.80 | 5306.43 | 40.24 | 63925.33 |

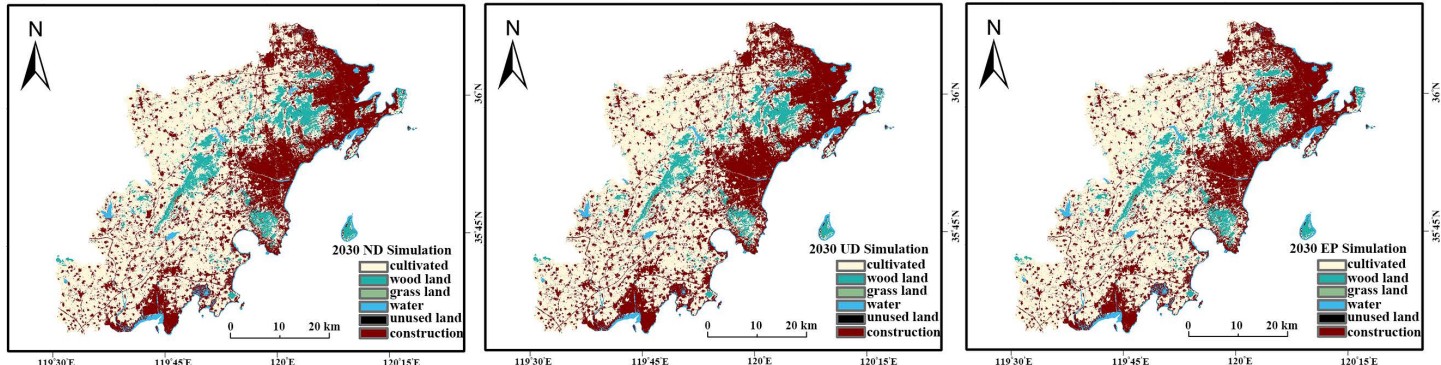

**Fig 12. Land Use Simulation Under Natural Development, Urban Development, and Ecological Protection Scenarios in 2030.**

economic growth, it also highlights a significant trade-off, with substantial impacts on valuable cultivated land and ecological spaces.

Ecological Protection (EP) Scenario: This scenario prioritizes ecological protection and restoration. The simulation results present a distinctly different pattern: construction land expansion is effectively controlled (projected to be the smallest area among the three scenarios at 63,925.33 ha). Cultivated land is best protected (reaching 122,677.38 ha), and the total area of forest land, grassland, and water bodies is optimized. This scenario is considered the model most aligned with the region's sustainable development goals.

**4.3.3. Future ecological sensitivity under different scenarios.** The simulated 2030 land use maps were used as inputs to predict the spatial distribution of ecological sensitivity for each scenario, assuming other evaluation factors remain constant (Fig 13).

The results indicate that the overall spatial distribution of ecological sensitivity will likely remain stable, with low-sensitivity areas concentrated in urban cores and high-sensitivity areas in the mountainous regions. However, the area proportions of each sensitivity level vary significantly across the scenarios.

To assess the potential benefits of an ecology-first strategy, we compared the 2020 sensitivity pattern with the 2030 EP scenario prediction (Table 11 vs Table 16) The comparison reveals that under the EP scenario, the area of moderately sensitive zones is projected to decrease slightly, while the areas of high and extremely high sensitivity zones continue to increase. This trend suggests that an ecological protection framework is projected to foster positive environmental changes, promoting the overall ecological balance of the West Coast New Area. These findings reinforce the critical role of proactive ecological zoning, consistent with recommendations for other ecologically vulnerable regions [6].

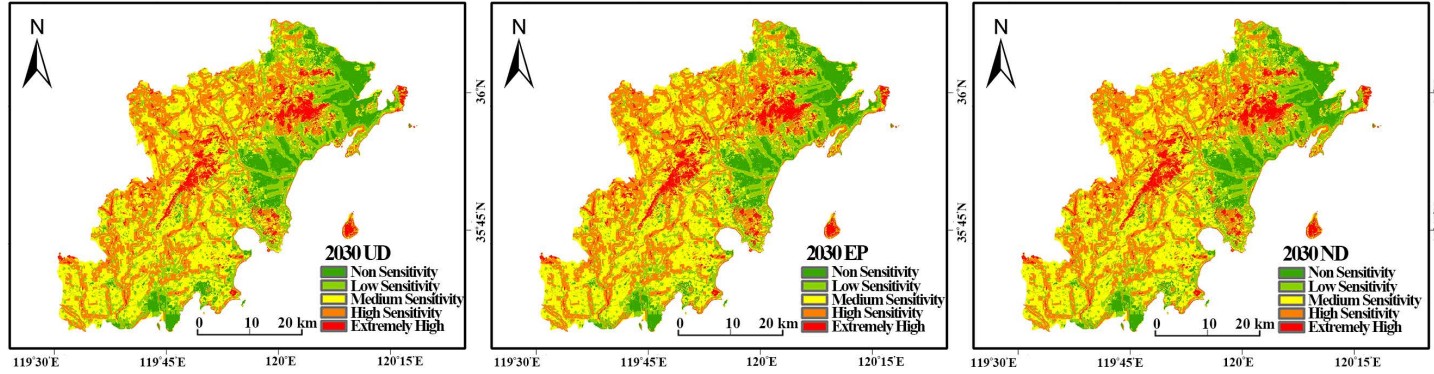

**Fig 13. Ecological Sensitivity Assessment Maps for Three Different Scenarios.**

**Table 16. Area and Percentage of Each Ecological Sensitivity Zone Under Three Different Scenarios (ha).**

| Time | ND | | UD | | EP | |
|---|---|---|---|---|---|---|
| | Area | % | Area | % | Area | % |
| non-sensitive | 22972.82 | 11.00% | 23994.63 | 11.49% | 22322.28 | 10.69% |
| low sensitivity | 37558.42 | 17.99% | 40177.78 | 19.24% | 35722.95 | 17.11% |
| moderate sensitivity | 74554.23 | 35.71% | 74272.43 | 35.57% | 75593.60 | 36.20% |
| high sensitivity | 58944.19 | 28.23% | 57250.84 | 27.42% | 60007.08 | 28.74% |
| extremely high sensitive | 14771.24 | 7.07% | 13105.21 | 6.28% | 15155.00 | 7.26% |
| Total | 208800.90 | 1 | 208800.90 | 1 | 208800.90 | 1 |

## 5. Policy implications

Based on these findings, we propose three specific policy recommendations for the QWCNA and similar coastal new areas. First, planning authorities should prioritize the establishment of 'green corridors' connecting the high-sensitivity zones of Xiaozhu, Dazhu, and Cangma-Tiejue mountains to enhance habitat connectivity and combat fragmentation. Second, the rapidly shrinking moderately sensitive zones, primarily agricultural land, should be designated as 'ecological buffer zones' with stricter development controls to prevent further urban encroachment. Third, future urban expansion should be directed exclusively towards infill development within the existing non-sensitive and low-sensitivity areas, promoting a more compact and efficient urban form consistent with the EP scenario's optimal land use structure.

## 6. Limitations and future research directions

Despite the robust findings of this study, several limitations remain, which also point to directions for future research:

First, the determination of factor weights in the ecological sensitivity assessment, while structured through an AHP-Delphi process, contains a degree of subjectivity. Although the Consistency Ratio (CR < 0.1) confirms the logical consistency of the judgments, future studies could employ more objective weighting methods or conduct a formal sensitivity analysis to test how variations in weights affect the final sensitivity patterns.

Second, while the PLUS model's integrated Random Forest algorithm is powerful for analyzing driving forces, future studies could benefit from an integrative approach that compares its performance with other machine learning models, such as Artificial Neural Networks (ANN) or Support Vector Machines (SVM), to explore the complex, non-linear relationships of land use change in even greater depth.

Third, while our PLUS model validation showed high accuracy for the historical period, all future simulations are subject to uncertainty. The scenarios are based on current policies and trends, which may shift over time. Future work could incorporate dynamic, adaptive scenarios that respond to potential policy or climate changes.

Fourth, our analysis of temporal change is based on the net area changes of sensitivity classes. While substantial, this does not include pixel-level statistical significance testing of the changes, which could provide finer-grained insights.

## 7. Conclusions

This study aimed to investigate the long-term spatiotemporal evolution characteristics of the ecological environment in the Qingdao West Coast New Area and predict future development trends to provide the scientific basis and decision-making support for regional ecological, environmental protection, and sustainable development. By integrating land use change analysis, ecological sensitivity assessment, and model simulations, this study drew the following main conclusions:

1) Significant Land Use Transformation (1990–2020): The West Coast New Area underwent profound land use changes over the past three decades. Driven by rapid urbanization, construction land expanded dramatically (a net increase of 31,575.15 ha), while cultivated land shrank significantly (a net decrease of 27,205.74 ha). This primary conflict reshaped the region's land use structure.

2) Spatial Polarization of Ecological Sensitivity (1990–2020): The region's ecological sensitivity became increasingly polarized. Lower-sensitivity areas, concentrated in the urbanized coastal zones, saw their internal structure shift as non-sensitive areas expanded (from 6.37% to 10.05%) at the expense of low-sensitivity areas. Moderate-sensitivity areas, mainly in the urban-rural transition zones, shrank dramatically (from 56.04% to 38.03%), representing the primary zone of land use conflict. Conversely, Higher-sensitivity areas, concentrated in the core ecological mountain systems, expanded significantly (from 18.04% to 36.50%). This expansion provides strong evidence that targeted ecological protection measures have achieved positive results.

3) The Ecological Protection (EP) Scenario as the Optimal Future Pathway: Simulations indicate that the EP scenario is most aligned with the region's sustainability goals and the 'Three Control Lines' management requirements. Under this scenario, the expansion of low-sensitivity (high-intensity development) areas is most effectively controlled, while the area of high-sensitivity (core ecological) zones is maximized. This demonstrates that an ecology-first strategy, which strictly adheres to urban development boundaries and ecological protection red lines, is the key to ensuring regional ecological security.

In summary, the Qingdao West Coast New Area exemplifies a landscape shaped by the dual forces of rapid urbanization and targeted conservation. Over the past three decades, this has resulted in a polarized ecological pattern: an expanding urban core and strengthening ecological barriers, at the expense of intermediate agricultural landscapes. Looking ahead, simulations based on the PLUS model confirm that a development pathway prioritizing ecological protection (the EP scenario) is not only viable but essential for achieving sustainable development. This approach effectively manages the sprawl of high-intensity development while enhancing the integrity of core ecological spaces, offering a clear and scientifically grounded strategy for the region's future.

## Supporting information

**S1 File.**
(DOCX)

## Acknowledgments

I am deeply grateful to all my colleagues in the research group for their help with experimental design, data analysis, and paper revisions. Their teamwork and encouragement played a crucial role in this research.

## Author contributions

**Conceptualization:** Tong Zhou, Jiabin Wang.

**Formal analysis:** Tong Zhou, Jiabin Wang.

**Methodology:** Tong Zhou, Jiabin Wang.

**Project administration:** Yaning Zhao, Yi Sheng.

**Software:** Jiabin Wang, Yaning Zhao, Yi Sheng.

**Validation:** Tong Zhou.

**Visualization:** Jiabin Wang, Yaning Zhao, Yi Sheng.

**Writing – original draft:** Jiabin Wang.

**Writing – review & editing:** Tong Zhou, Jiabin Wang.

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
