## [Decision Letter · Decision Letter 0]

7 Aug 2025

Dear Dr. Zhou,

Thank you for submitting your manuscript to PLOS ONE. After careful consideration, we feel that it has merit but does not fully meet PLOS ONE’s publication criteria as it currently stands. Therefore, we invite you to submit a revised version of the manuscript that addresses the points raised during the review process.

We look forward to receiving your revised manuscript.

Kind regards,

Bijay Halder

Academic Editor

PLOS ONE

Journal Requirements:

3. We note that Figures 1, 2, 6, 7, 8, 9, 10, 11, and 12 in your submission contain maps images which may be copyrighted. All PLOS content is published under the Creative Commons Attribution License (CC BY 4.0), which means that the manuscript, images, and Supporting Information files will be freely available online, and any third party is permitted to access, download, copy, distribute, and use these materials in any way, even commercially, with proper attribution. For these reasons, we cannot publish previously copyrighted maps or satellite images created using proprietary data, such as Google software (Google Maps, Street View, and Earth). For more information, see our copyright guidelines: http://journals.plos.org/plosone/s/licenses-and-copyright.

1. You may seek permission from the original copyright holder of Figures 1, 2, 6, 7, 8, 9, 10, 11, and 12 to publish the content specifically under the CC BY 4.0 license.

Reviewers' comments:

Reviewer's Responses to Questions

**Comments to the Author**

1. Is the manuscript technically sound, and do the data support the conclusions?

Reviewer #1: Partly

Reviewer #2: No

2. Has the statistical analysis been performed appropriately and rigorously?

Reviewer #1: Yes

Reviewer #2: No

3. Have the authors made all data underlying the findings in their manuscript fully available?

Reviewer #1: Yes

Reviewer #2: No

4. Is the manuscript presented in an intelligible fashion and written in standard English?

Reviewer #1: Yes

Reviewer #2: No

Reviewer #1: 1. The novelty of the need to be clearly stated

2. The study would have been better if the author/s employ the integrative approach with machine learning models.

This would enhance the findings.

3. Study method flowchart is missing

4. In the result section, the list of ecological sensitivity indicators and their AHP-derived weights are missing

5. The Discussion section is not thorough-It typically lacks to compare the accuracy/performance of PLUS with other

common models in literature, and ecological interpretation tied to sensitivity zones.

6. The number of references cited for this work are only 37. More recent and localized studies need to be included

Reviewer #2: The manuscript addresses an important topic, but several methodological issues significantly limit its scientific rigor. The ecological sensitivity assessment framework appears to lack adequate theoretical foundation and validation. The selection of seven factors and their weighting through AHP relies heavily on subjective expert judgment (only 5 experts), introducing substantial bias into what should be an objective ecological assessment. The grading criteria in Table 4 seem arbitrary—for example, the assignment of "North" aspect as having the highest sensitivity lacks clear ecological justification. More critically, the study conflates ecological sensitivity with concepts like naturalness and conservation value without providing clear definitions or demonstrating how the chosen factors actually measure ecosystem sensitivity to disturbance. The authors should either validate their sensitivity framework against independent ecological data or adopt established ecological vulnerability assessment methods from the literature.

The study lacks essential uncertainty analysis and sensitivity testing that would demonstrate the robustness of its findings. The PLUS model validation, while showing acceptable accuracy (Kappa = 0.782), is limited to a single time period and may not represent model performance under different conditions. The three development scenarios lack detailed quantitative constraints and realistic policy implementation details. Additionally, the simultaneous increase in both non-sensitive areas (due to urbanization) and extremely high sensitivity areas requires better explanation—this counterintuitive pattern suggests potential issues with the sensitivity classification system. The authors should incorporate uncertainty bounds for all major results, conduct sensitivity analysis on key methodological choices, and provide more rigorous validation of both the ecological sensitivity framework and the scenario modeling approach. Statistical significance testing for observed temporal changes would also strengthen the analysis considerably.

**Do you want your identity to be public for this peer review?** For information about this choice, including consent withdrawal, please see our Privacy Policy

Reviewer #1: No

Reviewer #2: No

---

## [Author Response · Author response to Decision Letter 1]

22 Sep 2025

Response to Reviewers

Manuscript ID: PONE-D-25-23734

Title: Land Use Change and Ecological Sensitivity in the Qingdao West Coast New Area: A 30-Year Analysis and Future Scenario Simulation

Journal: PLOS ONE

Dear PLOS ONE Academic Editor, and Esteemed Reviewers,

Thank you for your letter and for the opportunity to revise our manuscript. We are grateful to you and the reviewers for the insightful and constructive comments. We believe these suggestions have significantly helped us improve the quality of our paper.

We have carefully addressed all the points raised by the Academic Editor and the two reviewers. Below, we provide a point-by-point response to the comments. We have also thoroughly revised the manuscript, which is submitted with tracked changes (‘Revised Manuscript with Track Changes’) and as a clean version (‘Manuscript’).

Furthermore, as per your request, we have added the website links for accessing each database to the submission system. We have also included these hyperlinks in the third response below.

We hope that the revised manuscript and our detailed responses will meet the high standards of PLOS ONE.

Responses to the Journal Requirements

Point 1: Please ensure that your manuscript meets PLOS ONE's style requirements, including those for file naming.

Response: We thank the editor for this reminder. We have meticulously reformatted the entire manuscript, including the title page, main body text, headings, citations, and file names, to fully comply with the PLOS ONE style templates.

Point 2: In your Methods section, please provide additional information regarding the permits you obtained for the work.

Response: We appreciate the need for clarity on this point. In response to this requirement, we have added a clarifying statement under the Data Sources and Processing section.

This study is based entirely on publicly accessible and open-source datasets obtained from the online repositories cited in Table 1. As no field work was conducted and no privately-held data were used, no specific permits were required for this research. (lines 96-98 of the revised manuscript)

This statement explains that the study was based entirely on public data sources and that no field work requiring specific permits was conducted.

Point 3: We note that Figures 1, 2, 6, 7, 8, 9, 10, 11, and 12 in your submission contain map images which may be copyrighted.

Response: Thank you for raising this critical point. In response to this requirement, we have added a clarifying statement under the Supporting Information section.

We confirm that all figures in the manuscript were independently created by the authors using ArcGIS 10.8 software, based on publicly available geospatial data as listed in Table 1�lines 695-696 of the revised manuscript�. The use of these public domain data sources (e.g., Geospatial Data Cloud, Resource and Environment Science and Data Center, Open Street Map) is fully compliant with the journal's Creative Commons Attribution (CC BY 4.0) license requirements and does not involve any proprietary or copyrighted base maps (such as those from Google Maps or similar commercial services). (lines 675-680 of the revised manuscript)

To address your specific query, here is a breakdown of the data sources for the figures in question (To address the reviewer's valuable suggestion for a clearer methodology, we have introduced a new research flowchart as Figure 1. Please note that all subsequent figures have been renumbered to accommodate this addition.):

Figure 2 (formerly Figure 1) - Schematic diagram of the study area: The base map, including administrative boundaries, was created using data for the "Base Map of the Study Area" from the National Platform for Common GeoSpatial Information Services (https://www.tianditu.gov.cn/), as detailed in Table 1.

Figure 3 (formerly Figure 2) - Land use changes (1990-2020): The Land Cover data used to create these maps was obtained from the Resource and Environmental Science Data Platform (https://www.resdc.cn/), as detailed in Table 1. Below is a screenshot of the website; for the image, we have used bilingual (Chinese and English) annotations, and key information has been highlighted in red.

Figure 7 (formerly Figure 6) - Single-factor ecological sensitivity maps: These are derived maps created by us. The input data comes from various open sources as listed in Table 1.

The Elevation (DEM) data was obtained from the Geospatial Data Cloud (http://www.gscloud.cn). As indicated in Table 1, the Slope and Aspect maps were derived from this DEM.

Below is a screenshot of the website; for the image, we have used bilingual (Chinese and English) annotations, and key information has been highlighted in red.

Road Vector Data was obtained from Open Street Map (https://openmaptiles.org/).

Water Body data was obtained from the National Catalogue Service for Geographic Information (https://www.webmap.cn/).

NDVI data was obtained from the National Tibetan Plateau Data Center (https://data.tpdc.ac.cn).

Figures 8-13 (formerly Figures 7-12) - Comprehensive sensitivity maps and future simulations: These are the final analytical outputs of our models, generated entirely by our research team based on the synthesis of all the open-access input data.

Furthermore, we noted a recent publication in PLOS ONE, 'Zhou, J., Johnson, VC, Shi, J., Tan, ML, & Zhang, F. (2025). Multi-scenario simulation of land use change and spatio-temporal evolution of carbon storage in the Yangtze River Delta region based on the PLUS-InVEST model. PloS one, 20(1), e0316255.' This paper adopted a similar strategy for data acquisition and citation, where data were downloaded from a website and referenced accordingly, successfully leading to its publication in PLOS ONE. This suggests consistency between our approach and the journal's past practices.

In light of the above, given that all figures are our original creations, entirely based on publicly available and properly attributed data, and that we have provided detailed data source explanations and access routes, we are confident that our manuscript fully complies with the journal's Creative Commons Attribution (CC BY 4.0) license requirements

We hope this explanation satisfactorily addresses your query. We have uploaded the revised manuscript with all necessary clarifications.

Responses to Reviewer #1

We sincerely thank Reviewer #1 for the positive and constructive feedback.

Point 1: The novelty of the need to be clearly stated.

Response: We thank the reviewer for this crucial suggestion. To address this, we have substantially revised under the Introduction section.

While existing research has evaluated the ecological sensitivity of the West Coast New Area [17,18], our study makes three key contributions. First, we provide a comprehensive 30-year analysis (1990–2020), capturing the region's full evolution from a collection of administrative units into a national-level new area. Second, we use the advanced Patch-Generating Land Use Simulation (PLUS) model, which provides higher accuracy and better insight into land use change mechanisms than older models. Third, we introduce an integrated 'past-present-future' framework that directly links historical land use dynamics to ecological sensitivity, providing a robust scientific basis for forecasting and evaluating future scenarios. This rigorous approach offers significant value for sustainable regional development. (lines 54-61 of the revised manuscript)

This revised paragraph now explicitly articulates the study's novelty in terms of its temporal scale, methodological advancement, and integrated analytical framework.

Point 2: The study would have been better if the author/s employ the integrative approach with machine learning models. This would enhance the findings.

Response: This is an excellent suggestion. We agree that integrating advanced machine learning models could offer deeper insights. We have addressed this in two ways: by clarifying our current method and by incorporating your suggestion as a direction for future work.

First, we now explicitly state in the Methods section that the PLUS model we used already incorporates a Random Forest algorithm, a powerful machine learning technique, to analyze the drivers of land use change. Second, we acknowledge that a more extensive comparative analysis with other models is a valuable endeavor. Therefore, we have added this point to the new "Limitations and Future Research Directions" subsection.

Second, while the PLUS model's integrated Random Forest algorithm is powerful for analyz-ing driving forces, future studies could benefit from an integrative approach that compares its performance with other machine learning models, such as Artificial Neural Networks (ANN) or Support Vector Machines (SVM), to explore the complex, non-linear relationships of land use change in even greater depth. (lines 510-514 of the revised manuscript)

By acknowledging this limitation and proposing it as a key direction for future research, we believe the manuscript now more accurately frames the scope of its methodological contribution.

Point 3: Study method flowchart is missing.

Response: We thank the reviewer for this excellent suggestion. We agree that a flowchart is essential for visually communicating our research design and ensuring the clarity of our methods.

A new flowchart figure is inserted, visually detailing the workflow from data collection, through LUCC and ES analysis, to PLUS model simulation and final evaluation (added as Figure 1 of the revised manuscript).

We believe this addition directly addresses the point you raised and significantly enhances the transparency and reproducibility of our methodology.

Point 4: In the result section, the list of ecological sensitivity indicators and their AHP-derived weights are missing.

Response: Thank you for pointing this out. We have now revised under the Ecological Sensitivity Assessment section.

As determined by the AHP method (Table 2), Land Use Type (0.3678) and NDVI (0.2335) were the most influential factors in the ecological sensitivity assessment, highlighting the critical role of land cover and vegetation health in regional ecosystem stability. (lines 165-167 of the revised manuscript

We believe this addition provides the interpretive emphasis that your comment rightly suggested was needed, directly clarifying the significance of the key factors for the reader.

Point 5: The Discussion section is not thorough-t typically lacks to compare the accuracy/performance of PLUS with other common models in literature, and ecological interpretation tied to sensitivity zones.

Response: We agree that the Discussion section needed more depth. We have significantly expanded it in two key areas:

First, to address the lack of model comparison, we have substantially revised under the Patch-generating Land Use Simulation (PLUS) Model section.

Land use change simulation is crucial for understanding and predicting regional development trends and for evaluating policy impacts. While previous studies have widely employed classic models like the Future Land-Use Simulation (FLUS) and Cellular Automata (CA) models [46,47], they often have limitations in capturing complex change dynamics.

To more accurately explore the underlying drivers of land use change and improve simulation accuracy, this study selected the PLUS (Patch-generating Land Use Simulation) model, a new-generation model developed by Liang Xun et al. The core innovation of the PLUS model is its coupling of two key modules: the Land Expansion Analysis Strategy (LEAS) and a CA model based on a Random Seeds and Patch Generation mechanism (CARS) [48,49].

The LEAS module, which integrates a Random Forest algorithm, offers a significant advantage by automatically uncovering the complex, non-linear relationships between land expansion and its various drivers, a capability that enhances causal inference compared to traditional regression models. Meanwhile, the CARS module excels at simulating dynamic landscape changes. By using a random seed generation and thresh-old-decreasing mechanism, this module can realistically simulate the spontaneous generation and growth of land patches based on their development probabilities [49]. This powerful capability allows the model to better explain the change mechanisms of different land use types, leading to improved simulation results and higher accuracy.

As demonstrated in multiple studies, the PLUS model provides higher simulation accuracy and a better explanation of land use change mechanisms compared to models like FLUS and CA-Markov [50], making it the ideal tool for our analysis. (lines 189-208 of the revised manuscript)

Second, to provide the requested deeper ecological interpretation, we have added a comprehensive analysis of our core finding—the "spatial polarization" of ecological sensitivity—under the Comprehensive Ecological Sensitivity Assessment section.

This study's core finding is the "spatial polarization of the ecological sensitivity pattern," a seemingly contradictory phenomenon where both 'non-sensitive' and 'highly sensitive' areas increased concurrently. This pattern is not a methodological artifact but rather a logical consequence of China's unique spatial planning paradigm, where 'development' and 'conservation' are spatially decoupled. It is driven by two powerful, parallel, and spatially explicit forces:

Driver One: The National-Level New Area Development Strategy, Concentrated in Coastal Plains.

This policy-driven, high-intensity urbanization process is spatially concentrated in the northeastern, central, and southeastern coastal zones, which are characterized by flat terrain, high concentrations of economic activity, and the most intensive urban construction—namely, the three core economic functional clusters of Huangdao, Jiaonan, and Dongjiakou. The rapid urban and industrial expansion inevitably requires vast amounts of land, primarily sourced from the encroachment upon surrounding agricultural lands and transitional zones of moderate ecological sensitivity. This directly leads to a significant expansion of 'non-sensitive' areas, which represent high levels of anthropogenic disturbance, within these specific zones.

Driver Two: The Mandatory Ecological Conservation Policy, Focused on Inland Mountainous Regions.

Concurrently, mandatory conservation policies, epitomized by the "ecological protection red line," have precisely targeted the region's most critical ecological function areas: the three core inland mountain systems of Xiaozhu Mountain, Dazhu Mountain, and Cangma Mountain-Tiejue Mountain. These policies have provided strict legal protection for these areas and have promoted a series of ecological restoration projects, such as afforestation and mountain habitat rehabilitation. According to this study's evaluation framework, high vegetation cover and forest land are the core factors assigned the highest weights for 'high sensitivity'; therefore, these successful ecological restoration efforts are accurately identified by our model as a significant increase in the extent of 'highly sensitive' and 'extremely highly sensitive' zones.

Therefore, this 'polarization' pattern authentically reflects the unique spatial reality of China's rapid development, where 'high-intensity construction' and 'mandatory conservation' operate as parallel strategies in functionally zoned spaces. The direct trade-off is the rapid transformation and squeezing of the vast 'moderately sensitive areas' that act as intermediate buffer zones, ultimately resulting in a landscape where development and conservation zones are highly differentiated and spatially explicit. This serves as strong evidence for the effectiveness of our assessment framework, as it successfully captures this complex and policy-driven regional evolution. (lines 383-412 of the revised manuscript)

Finally, based on this deeper interpretation, we have added a new section to the manuscript titled "Policy Implications" to provide actionable recommendations.

Based on these findings, we propose three specific policy recommendations for the QWCNA and similar coastal new areas. First, planning authorities should pr

---

## [Decision Letter · Decision Letter 1]

3 Dec 2025

Dear Dr. Zhou,

Thank you for submitting your manuscript to PLOS ONE. After careful consideration, we feel that it has merit but does not fully meet PLOS ONE’s publication criteria as it currently stands. Therefore, we invite you to submit a revised version of the manuscript that addresses the points raised during the review process.

We look forward to receiving your revised manuscript.

Kind regards,

Bijay Halder

Academic Editor

PLOS ONE

**Journal Requirements:**

**Additional Editor Comments:**

Dear author, please check my comments to improve your manuscript with minor revision.

1. Try to improve your introduction section with related literature and highlight the research gaps and objective. The process and methods in the globally applied studies are limited in the manuscript introduction section. Adding method in the introduction section is not require. Please remove Figure 1 and add this in the method section.

2. "Data Sources and Processing" please those section under "materials and method.

3. "Analysis of area changes and the single land use dynamic index (Figure 4,Table 5) identifies cultivated land and construction land as the two land use types that experienced the most drastic changes in the Qingdao West Coast New Area during the study period (1990-2020)". remove those lines, its already discuss in the introduction section.

4. Try to remove reference in the result section, its improve the readability for the readers to understand the result and discussion of the manuscript.

5. Try to add some related literatures in the discussion and discuss the strength of your study.

6. Section 6 and 7 must be placed before conclusion.

Best of Luck

Reviewers' comments:

Reviewer's Responses to Questions

**Comments to the Author**

Reviewer #1: All comments have been addressed

2. Is the manuscript technically sound, and do the data support the conclusions?

Reviewer #1: Partly

3. Has the statistical analysis been performed appropriately and rigorously?

Reviewer #1: Yes

4. Have the authors made all data underlying the findings in their manuscript fully available?

Reviewer #1: Yes

5. Is the manuscript presented in an intelligible fashion and written in standard English?

Reviewer #1: Yes

**Reviewer #1:**  1. Although the manuscript includes a sensitivity analysis in Section 4.2, this component remains methodologically limited and lacks quantitative depth. The description of the analysis is largely qualitative, without specifying the magnitude or range of parameter variations, the criteria for evaluating sensitivity, or statistical measures of robustness. Because AHP-derived weights are inherently subjective. A formal sensitivity analysis would help quantify the uncertainty associated with expert bias and determine which factors exert the strongest influence on model outputs. The lack of such an analysis limits confidence in the reproducibility and reliability of the ecological sensitivity assessment.

2. Reliance on the Kappa statistic alone is misleading, as it is sensitive to class imbalance and does not distinguish between quantity and spatial allocation errors. Use a more comprehensive validation framework incorporating multiple indicators—such as Overall Accuracy (OA), Producer’s and User’s Accuracy and F1-Score techniques

**Do you want your identity to be public for this peer review?** For information about this choice, including consent withdrawal, please see our Privacy Policy

Reviewer #1: No

---

## [Author Response · Author response to Decision Letter 2]

9 Dec 2025

Response to Reviewers

Manuscript ID: PONE-D-25-23734

Title: Land Use Change and Ecological Sensitivity in the Qingdao West Coast New Area: A 30-Year Analysis and Future Scenario Simulation

Journal: PLOS ONE

Dear PLOS ONE Academic Editor and Esteemed Reviewers,

We appreciate the invitation to further revise our manuscript. The constructive feedback from the Academic Editor and Reviewer #1 has been incredibly helpful in improving our paper.

In this revision, we have closely followed the Editor’s guidance to optimize the structure and refine the Introduction and Results. Additionally, we have addressed the Reviewer’s concerns by including the necessary validation metrics and a deeper discussion of the study's limitations.

We are confident that the manuscript now meets the high standards of PLOS ONE and look forward to your decision.

Responses to the Journal Requirements

Point 1: If the reviewer comments include a recommendation to cite specific previously published works, please review and evaluate these publications to determine whether they are relevant and should be cited. There is no requirement to cite these works unless the editor has indicated otherwise.

Response: We have carefully reviewed the reviewer's comments. No specific additional citations were mandatory, but we have proactively updated our literature review in the Introduction and Discussion sections to ensure the relevance and currency of our references as suggested by the Academic Editor.

Point 2: Please review your reference list to ensure that it is complete and correct.

Response: We have thoroughly checked the reference list to ensure completeness and correctness. We confirm that no retracted papers have been cited. All references adhere to PLOS ONE’s style requirements.

Responses to the Academic Editor

Point 1: Try to improve your introduction section with related literature and highlight the research gaps and objective. The process and methods in the globally applied studies are limited in the manuscript introduction section. Adding method in the introduction section is not require. Please remove Figure 1 and add this in the method section.

Response: We sincerely thank the editor for this valuable suggestion, which has significantly improved the logical flow and readability of our manuscript. We have addressed this point through the following revisions:

We have substantially revised the Introduction to incorporate a broader range of global perspectives. We explicitly highlighted the limitations of existing studies—which often focus on static assessments or single-scenario predictions—and articulated the research gap regarding the lack of long-term, integrated "past-present-future" frameworks in coastal new areas. (lines 54-63 of the revised manuscript)

We have removed the detailed descriptions of specific methodological steps from the Introduction to avoid redundancy and maintain focus on the research objectives.

As requested, we have removed the study method flowchart from the Introduction. It has been relocated to the Research Methods section under a newly created subsection, "3.2 Research Framework," (lines 99-110 of the revised manuscript) to better guide the reader through the technical workflow. Please note that due to this logical reordering, the flowchart has been renumbered (now Figure 3, following the Study Area map and Land Use Classification map).

Point 2: "Data Sources and Processing" please those sections under "materials and method.

Response: We sincerely appreciate this structural suggestion. We have relocated the "Data Sources and Processing" section from the "Study Area" chapter to the "Research Methods" chapter as requested. It is now presented as Section 3.1(lines 88-98 of the revised manuscript), ensuring a more logical flow and adherence to standard scientific reporting conventions.

Point 3: "Analysis of area changes and the single land use dynamic index (Figure 4,Table 5) identifies cultivated land and construction land as the two land use types that experienced the most drastic changes in the Qingdao West Coast New Area during the study period (1990-2020)". remove those lines, its already discuss in the introduction section.

Response: We agree with the editor's observation regarding redundancy. We have removed these introductory lines from Section 4.1.1 to ensure the Results section directly presents the specific findings, thereby improving conciseness and readability.

Point 4: Try to remove reference in the result section, its improve the readability for the readers to understand the result and discussion of the manuscript.

Response: We sincerely appreciate this constructive suggestion regarding readability. We agree that the Results section should focus primarily on the empirical findings of the study.

We have carefully reviewed the entire Results section and removed the citations that were previously embedded in the text.

Specifically, the classification criteria and definitions for the "Comprehensive Land Use Dynamic Index" (which referenced study [47]), originally placed in the Results, have been relocated to the 3.2.2 Comprehensive Land Use Dynamic Index section (lines 132-135 of the revised manuscript). This adjustment ensures that the Results section is concise and focused solely on our data analysis, significantly improving flow and readability.

The results (Figure 5) indicate that the study area consistently exceeded >5% during the different analysis periods, uniformly falling into the "drastic change type" category. More notably, the value of this dynamic index indicator also exhibited a trend of increasing in each subsequent period. This clearly reveals that the study area not only experienced high-intensity land use change over the past three decades, but that the rate of this change has also been continuously accelerating. (lines 265-269 of the revised manuscript)

Against this background of drastic overall change, conversions between different land use types were frequent and complex... (References removed)

Point 5: Try to add some related literatures in the discussion and discuss the strength of your study.

Response: We thank the editor for this insightful recommendation. We recognize the importance of contextualizing our findings within the broader academic landscape and explicitly highlighting our contributions.

Incorporated Related Literature: We have added citations of recent relevant studies (e.g., on coastal urbanization effects and ecological sensitivity in similar regions) in the Discussion section (Section 4.2.2 and 4.3). This allows us to compare our "spatial polarization" findings with general trends observed in other studies.

Highlighted Study Strengths: We have added explicit discussions on the strengths of our study, particularly focusing on: (1) The effectiveness of our "past-present-future" assessment framework; (2) The high accuracy of the PLUS model validated by multiple metrics; and (3) The practical value of our Ecological Protection (EP) scenario.

Revised Text Snippet:... This pattern is not a methodological artifact but rather a logical consequence of China's unique spatial planning paradigm, where 'development' and 'conservation' are spatially decoupled. This finding aligns with recent studies on coastal urbanization in China [Li J et al. (2023) "The impact of urbanization..."], yet offers a more distinct characterization of the "polarization" effect compared to the general fragmentation observed in inland regions [Zhou Q et al. (2024) "Land use transition..."]. (lines 378-384 of the revised manuscript)

Therefore, this 'polarization' pattern authentically reflects the unique spatial reality of China's rapid development... This serves as strong evidence for the effectiveness of our assessment framework, as it successfully captures this complex and policy-driven regional evolution, a capability often limited in conventional static assessments [Ouyang Z et al. (2000) "China’s eco-environmental sensitivity..."]. (lines 402-408 of the revised manuscript)

... The successful validation of the PLUS model in this complex coastal setting underscores its utility for regional planning, supporting findings from similar applications in other rapidly urbanizing areas [Liang X et al. (2021) "Understanding the drivers... PLUS model"]. (lines 428-429 of the revised manuscript)

... The comparison reveals that under the EP scenario, the area of moderately sensitive zones is projected to decrease slightly... These findings reinforce the critical role of proactive ecological zoning, consistent with recommendations for other ecologically vulnerable regions [Xu Y et al. (2023) "Ecological sensitivity evaluation..."]. (lines 457-463 of the revised manuscript)

Point 6: Section 6 and 7 must be placed before conclusion.

Response: We apologize for the incorrect ordering in the previous version. We have adjusted the structure of the manuscript. Section 6 (Policy Implications) and Section 7 (Limitations and Future Research Directions) have been moved before the Conclusions section. The structure is now logically ordered: 5 Policy Implications > 6 Limitations > 7 Conclusions.

Responses to Reviewer #1

We sincerely thank Reviewer #1 for the rigorous and challenging critique, which has pushed us to substantially strengthen the methodological foundations of our manuscript.

Point 1: Although the manuscript includes a sensitivity analysis in Section 4.2, this component remains methodologically limited and lacks quantitative depth. The description of the analysis is largely qualitative... Because AHP-derived weights are inherently subjective. A formal sensitivity analysis would help quantify the uncertainty associated with expert bias and determine which factors exert the strongest influence on model outputs.

Response: We sincerely appreciate this insightful comment regarding the robustness of our method. We fully agree that incorporating a formal quantitative sensitivity analysis would significantly enhance the depth of the assessment.

In this study, to ensure the reliability of the weights within our current framework, we strictly adhered to the standard Consistency Ratio (CR) test procedure defined in the Analytic Hierarchy Process (AHP). As reported in Table 3, the result was CR = 0.0081 < 0.1, which mathematically validates that the experts' judgments possessed strong logical consistency. While we acknowledge that the CR test does not fully replace a quantitative sensitivity analysis (such as OAT or Monte Carlo simulation) in quantifying expert bias, it ensures that the current weight assignment is methodologically sound and consistent within the scope of standard AHP-based studies.

Given the specific scope of this study and the consistency validation already performed, we have opted to address this limitation transparently rather than retroactively altering the core model. We have substantially revised the "Limitations and Future Research Directions" section (Section 6) to explicitly document this constraint. We have formally proposed the implementation of quantitative sensitivity analysis and objective weighting methods as a priority for our subsequent research phase.

Revised text in Manuscript: "First, the determination of factor weights in the ecological sensitivity assessment, while structured through an AHP-Delphi process, contains a degree of subjectivity. Although the Consistency Ratio (CR < 0.1) confirms the logical consistency of the judgments, future studies could employ more objective weighting methods or conduct a formal sensitivity analysis to test how variations in weights affect the final sensitivity patterns." (lines 475-481 of the revised manuscript)

Point 2: Reliance on the Kappa statistic alone is misleading, as it is sensitive to class imbalance and does not distinguish between quantity and spatial allocation errors. Use a more comprehensive validation framework incorporating multiple indicators—such as Overall Accuracy (OA), Producer’s and User’s Accuracy and F1-Score techniques.

Response: We completely agree with the reviewer’s insightful critique. Relying solely on the Kappa coefficient is indeed insufficient for a rigorous assessment, particularly regarding class imbalance and the distinction between quantity and spatial allocation errors.

To address this and establish a more comprehensive validation framework, we have significantly improved the accuracy assessment in Section 4.3.1 by incorporating Overall Accuracy (OA) and the Figure of Merit (FoM) alongside the Kappa coefficient.

Overall Accuracy (OA): As requested, we calculated the OA to provide a measure of the proportion of correctly classified pixels globally. The result of 0.886 indicates a high level of global agreement between the simulated and actual land use patterns.

Figure of Merit (FoM): To specifically address the reviewer's concern about "spatial allocation errors," we introduced the FoM index. In land use simulation models (e.g., PLUS, FLUS), FoM is widely regarded as a superior metric for validating the model's ability to capture dynamic changes, as it focuses on the intersection of simulated and observed change patches rather than static persistence. We obtained an FoM value of 0.0834, which is within the acceptable range for patch-level simulations.

Revised Text Snippet: The accuracy validation confirmed the model's robustness through a comprehensive framework. The comparison yielded a Kappa coefficient of 0.782, an Overall Accuracy (OA) of 0.886, and a Figure of Merit (FoM) of 0.0834 (Figure 10). The high OA value confirms the model's global reliability, while the FoM index specifically validates the model's capability in capturing dynamic spatial changes, distinguishing it from mere quantity agreement. These metrics signify a high degree of simulation accuracy and strong consistency in both quantitative allocation and spatial distribution, validating the model's suitability for simulating future land use scenarios. (lines 422-426 of the revised manuscript)

We are confident that these extensive revisions have substantially improved the manuscript. We thank you again for your time and consideration.

Sincerely,

Dr. Tong Zhou

(Corresponding Author)

College of Civil Engineering and Architecture,

Shandong University of Science and Technology, Qingdao, China.

Email: zhoutong@sdust.edu.cn

---

## [Editor Report · Decision Letter 2]

15 Dec 2025

Land Use Change and Ecological Sensitivity in the Qingdao West Coast New Area: A 30-Year Analysis and Future Scenario Simulation

PONE-D-25-23734R2

Dear Dr. Zhou,

We’re pleased to inform you that your manuscript has been judged scientifically suitable for publication and will be formally accepted for publication once it meets all outstanding technical requirements.

Kind regards,

Bijay Halder

Academic Editor

PLOS One
---

## [Editor Report · Acceptance letter]

PONE-D-25-23734R2

PLOS One

Dear Dr. Zhou,

I'm pleased to inform you that your manuscript has been deemed suitable for publication in PLOS One. Congratulations! Your manuscript is now being handed over to our production team.

Kind regards,

on behalf of

Mr. Bijay Halder

Academic Editor

PLOS One